# ENHANCING PHYSICAL PLAUSIBILITY IN VIDEO GENERATION BY REASONING THE IMPLAUSIBILITY

## ABSTRACT

Diffusion models can generate realistic videos, but existing methods rely on implicitly learning physical reasoning from large-scale text-video datasets, which is costly, difficult to scale, and still prone to producing implausible motions that violate fundamental physical laws. We introduce a training-free framework that improves physical plausibility at inference time by explicitly reasoning about implausibility and guiding the generation away from it. Specifically, we employ a lightweight physics-aware reasoning pipeline to construct counterfactual prompts that deliberately encode physics-violating behaviors. Then, we propose a novel *Synchronized Decoupled Guidance* (SDG) strategy, which leverages these prompts through synchronized directional normalization to counteract lagged suppression and trajectory-decoupled denoising to mitigate cumulative trajectory bias, ensuring that implausible content is suppressed immediately and consistently throughout denoising. Experiments across different physical domains show that our approach substantially enhances physical fidelity while maintaining photorealism, despite requiring no additional training. Ablation studies confirm the complementary effectiveness of both the physics-aware reasoning component and SDG. In particular, the aforementioned two designs of SDG are also individually validated to contribute critically to the suppression of implausible content and the overall gains in physical plausibility. This establishes a new and plug-and-play physics-aware paradigm for video generation.

## 1 INTRODUCTION

Recent text-to-video diffusion models (Wan et al., 2025; Yang et al., 2025b) produce strikingly realistic sequences across diverse visual concepts and prompts. Yet despite impressive progress in fidelity and prompt adherence, their behavior often departs from everyday physics: objects accelerate without cause, fluids ignore gravity, and phase transitions misfire. These failure modes matter because if video generative models are to serve as general-purpose world simulators (Liu et al., 2025), they must respect physical commonsense, not merely aesthetics.

Emerging benchmarks explicitly validate such physical plausibility. For example, PhyGen-Bench (Meng et al., 2024) curates 160 prompts spanning 27 physical laws across four domains (mechanics, optics, thermal, and material properties) and introduces an automated evaluator. In parallel, VideoPhy (Bansal et al., 2025) evaluates real-world actions with fine-grained human judgments over semantic adherence, physical commonsense, and grounded physical-rule violations. Together, these studies show that current models frequently violate physical commonsense and that scaling or prompt-engineering alone does not solve the problem, highlighting a persistent gap that current video generation models can render, but struggle to reason physically.

Our core idea is to enhance physical plausibility by reasoning about implausibility, then guiding the video generation away from it. Specifically, motivated by the fact that user prompts are typically underspecified with respect to entities, scene conditions, interactions, and expected causal evolution, we first leverage a LLM-empowered physics-aware reasoning (PAR) pipeline to infer a physically valid trajectory, and construct a targeted counterfactual that violates the governing physical law while remaining visually plausible. To guide generations away from these counterfactuals, a naive way is to use negative prompting, but for which we identify two core gaps that limit the effectiveness, i.e., *lagged suppression effect* and *cumulative trajectory bias*. We therefore propose a novel Synchronized

Decoupled Guidance (SDG) approach with two designs (synchronized directional normalization and trajectory-decoupled denoising) that directly address these gaps. Notably, SDG serves as a plug-and-play inference-time strategy that requires no retraining or finetuning.

Extensive experiments validate that our framework maintains photorealism while improving physics-related scores across different physical phenomena such as solid mechanics, fluid dynamics, optics, and thermodynamics. On the PhyGenBench and VideoPhy benchmarks, we achieve consistent gains over strong base models such as CogVideoX-5B and Wan2.1-14B, and remain competitive with several physics-aware approaches that are not training-free. Ablation studies confirm that both components are necessary; PAR provides targeted, physics-aware counterfactuals, while SDG, via its two designs, turns those signals into non-delayed, unbiased suppression of implausible content.

In summary, our contributions are threefold. *First*, we introduce a reason-then-guide framework for physics-aware video generation that is training-free and model-agnostic. *Second*, we propose Synchronized Decoupled Guidance (SDG) with synchronized directional normalization and trajectory-decoupled denoising, addressing the lagged suppression and cumulative trajectory bias of negative prompting. *Third*, we demonstrate improvements on physics-focused benchmarks and validate through ablations the complementary roles of PAR and SDG. Taken together, our findings complement and extend the evidence from recent benchmarks that current video models need explicit physics-aware control to approach physically plausible generation.

## 2 PRELIMINARIES

### 2.1 CLASSIFIER-FREE GUIDANCE IN DIFFUSION MODELS

Diffusion models (Ho et al., 2020; Nichol & Dhariwal, 2021; Song et al., 2021) have emerged as a powerful family of generative methods. They define a forward process where Gaussian noise is progressively injected into a clean data $x_0$ over $T$ timesteps, yielding a fully noised sample $x_T \sim \mathcal{N}(0, \mathbf{I})$. The forward dynamics are expressed as:

$$q(x_t|x_{t-1}) = \mathcal{N}(x_t; \sqrt{1 - \beta_t}x_{t-1}, \beta_t\mathbf{I}), \tag{1}$$

where $x_t$ denotes the corrupted data at step $t$, and $\beta_{t=1}^T$ specifies the variance schedule controlling the noise level. To generate data, one trains a reverse denoising process that recovers $x_{t-1}$ from $x_t$. A neural network parameterized by $\theta$ is used to approximate the conditional distribution:

$$p_\theta(x_{t-1}|x_t) = \mathcal{N}(x_{t-1}; \mu_\theta(x_t), \sigma_\theta^2(x_t)\mathbf{I}). \tag{2}$$

In practice, the model $\epsilon_\theta(x_t, c, t)$ can be trained to predict the additive noise $\epsilon_t$ at each step, conditioned on side information $c$ such as a text prompt, rather than reconstructing $x_{t-1}$ directly. To better control the quality and relevance of generated samples, classifier-free guidance (CFG) (Ho & Salimans, 2022) is commonly adopted. CFG modifies the predicted noise at inference time by interpolating between the unconditional estimate $\epsilon_\theta(x_t, \emptyset, t)$ and the conditional estimate $\epsilon_\theta(x_t, c, t)$. Using a guidance strength $w > 1$, the final adjusted prediction is:

$$\hat{\epsilon}_t \leftarrow \epsilon_\theta(x_t, \emptyset, t) + w \cdot (\epsilon_\theta(x_t, c, t) - \epsilon_\theta(x_t, \emptyset, t)). \tag{3}$$

This simple mechanism provides a tunable trade-off between sample fidelity and diversity. Once the guided noise prediction $\hat{\epsilon}_t$ is obtained, the state update from $x_t$ to $x_{t-1}$ can be performed using a generic update rule that leverages $\hat{\epsilon}_t$:

$$x_{t-1} = \alpha_t x_t + \beta_t \hat{\epsilon}_t + \eta_t, \tag{4}$$

where $\alpha_t$ and $\beta_t$ are coefficients determined by the sampler, and $\eta_t$ represents optional stochasticity.

### 2.2 NEGATIVE PROMPTING

Negative prompting was first introduced in the Stable Diffusion 2.0 release and has since become a widely used technique for improving controllability in diffusion models (Stability AI, 2022; Woolf, 2023). The central idea is to not only specify desirable attributes through a positive prompt $p_+$, but also to explicitly provide a negative prompt $p_-$ that encodes features the model should avoid.

In contrast to classifier-free guidance (CFG), which interpolates between unconditional and conditional predictions (Eq. 3), negative prompting can be interpreted as anchoring the prediction on the

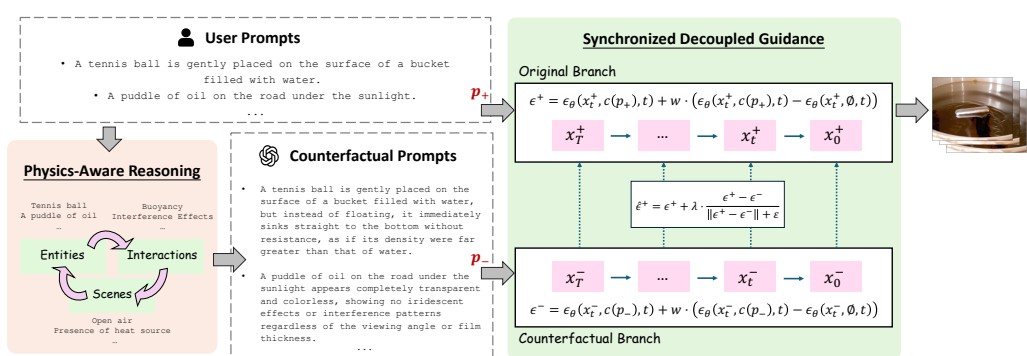

Figure 1: Overall framework. **Left: Physics-Aware Reasoning (PAR).** Given a user prompt, an LLM identifies entities, interactions, and scene conditions to produce a structured analysis of the underlying physical process. Based on this reasoning, it constructs counterfactual prompts that preserve the same entities and scenes but deliberately violate the governing physical law, yielding targeted physics-aware negatives. **Right: Synchronized Decoupled Guidance (SDG).** During denoising, we evolve two branches conditioned on the user prompt and the counterfactual prompt, respectively. Their noise estimates are combined with directional normalization and trajectory decoupling, ensuring that implausible structures are suppressed immediately and consistently throughout generation.

positive prompt while pushing it away from the negative prompt. This yields the following adjusted noise estimate (Armandpour et al., 2023):

$$\hat{\epsilon}_t \; \leftarrow \; \epsilon_\theta(x_t, c(p_+), t) \; + \; w \cdot \big( \epsilon_\theta(x_t, c(p_+), t) - \epsilon_\theta(x_t, c(p_-), t) \big), \tag{5}$$

where $c(p_+)$ is the embedding of the positive prompt (i.e., user prompt), $c(p_-)$ is the embedding of the negative prompt, and $w > 0$ controls the strength of suppression.

## 3 METHODOLOGY

Our methodology is built on two key components: (i) the construction of counterfactual prompts that deliberately invoke physically implausible behaviors, and (ii) a new guidance mechanism that leverages these prompts to enforce physics-awareness during video generation. Together, these components allow us to systematically expose and suppress violations of physical laws, without requiring retraining of the underlying diffusion model. The remainder of this section is organized as follows: Sec. 3.1 describes how counterfactual prompts are generated using a physics-aware reasoning pipeline. Sec. 3.2 then analyzes why naively incorporating these prompts through existing negative prompting remains insufficient, identifying two fundamental gaps. Finally, Sec. 3.3 presents our proposed *Synchronized Decoupled Guidance* (SDG), which integrates synchronized directional normalization and trajectory-decoupled denoising to directly address these gaps and fully exploit the counterfactuals for physics-aware generation.

### 3.1 PHYSICS-AWARE REASONING FOR COUNTERFACTUAL CONSTRUCTION

**The pipeline.** As shown in Fig. 1 (Left), we design a lightweight physics-aware reasoning pipeline powered by a large language model (LLM) to generate structured counterfactual prompts. Given a user prompt, the LLM performs two steps. First, in **physics reasoning**, it identifies relevant attributes such as entities, interactions, and environmental conditions, and infers the temporal evolution of the process, yielding a structured description of how the event would normally unfold under physical laws. Second, in **counterfactual construction**, it synthesizes a variant of the event that preserves the same entities and scene but deliberately violates the expected causal chain (e.g., the absence of bubbling when acid and base are mixed, or an object sinking instead of floating). These counterfactuals remain visually plausible yet physically implausible, providing targeted signals for subsequent guidance.

**The need for physics reasoning.** User-provided prompts are typically underspecified, often describing the surface-level visual content without reference to the underlying physical processes. For example, a prompt such as "A timelapse captures the transformation as water vapor in a humid environment comes into contact with a cool glass surface" specifies the entities (water vapor, glass surface) but omits the expected physical phenomenon and its outcome. Reasoning is therefore essential to enrich these prompts with the missing physical context, ensuring they become physically well-specified and suitable for constructing meaningful counterfactuals.

**The needs for constructing targeted and structured counterfacturals.** Our central idea is to improve physical plausibility by explicitly reasoning about implausibility. Such implausibility is informed by the constructed counterfactual prompt, which is leveraged by our proposed synchronized decoupled guidance (SDG) strategy (Sec. 3.3) to consistently steer the generation away from such implausible outcomes. To achieve effective guidance, we need the constructed counterfacturals to be targeted and structured.

Using the aforementioned user prompt as an example, under normal physical laws, this situation is governed by the principle of condensation: as warm vapor meets the cooler surface, the vapor cools to its dew point, releases latent heat, and gradually forms liquid droplets that coalesce and drip down. A meaningful counterfactual prompt in this case should therefore deliberately *violate condensation dynamics*, for instance by describing the surface as being covered with droplets instantly from the start, without any observable phase transition. However, when there is no explicit physics-aware reasoning about which entities interact, in what scene, and under which governing principles, counterfactual prompts risk violating irrelevant or unintended laws. For example, as shown in our ablation (Fig. 14), without physics-aware reasoning, the generation may result in a counterfactual such as "the vapor instantly freezes into solid ice upon contact," which is implausible in this context and fails to violate the expected condensation law. Instead of targeting the intended physical principle, such generic counterfacturals introduce unrelated violations.

To ensure that counterfactuals consistently target the correct physical law, we construct structured counterfactuals through physics-aware reasoning. By explicitly reasoning about entities, interactions, and scene conditions and keeping the entities and scene context unchanged, we can generate counterfactuals that remain visually plausible yet deliberately contradict the governing laws of the process, thereby providing effective signals for our guidance strategy (Sec. 3.3). Using the same example, our constructed counterfactual is "The glass surface is instantly covered in water droplets from the beginning, without any observable condensation or gradual droplet formation." Unlike the generic counterfacturals that introduce unrelated violations, this counterfactual directly contradicts the governing condensation law while preserving the same entities and scene context as the original prompt. By doing so, it avoids drifting into irrelevant outcomes and instead provides a targeted violation that the guidance mechanism can consistently suppress.

### 3.2 GAPS IN EXISTING NEGATIVE PROMPT GUIDANCE

A naive way to leverage our constructed counterfactuals for addressing the physical implausibility challenge is through the use of negative prompting (Woolf, 2023; Armandpour et al., 2023), which has proven useful for suppressing undesired semantics. However, we find its effectiveness to be inherently limited by the technique it is integrated into the CFG from the following two perspectives.

**Lagged Suppression Effect.** Eq. 5 shows that negative prompting modifies the predicted noise by subtracting a weighted discrepancy between the positive condition $c(p_+)$ (i.e., conditioned on the original user prompt) and negative condition $c(p_-)$ (i.e., conditioned on the undesired prompt, and in our case, this will be our constructed counterfactual prompts). Let the discrepancy vector at time $t$ be:

$$\Delta_t = \epsilon_\theta(x_t, c(p_+), t) - \epsilon_\theta(x_t, c(p_-), t). \tag{6}$$

Rewriting the equation for negative prompting (Eq. 5) gives:

$$\hat{\epsilon}_t \leftarrow \epsilon_\theta(x_t, c(p_+), t) + w \cdot \Delta_t, \tag{7}$$

where $w \cdot \Delta_t$ contains the suppression effect from negative prompting. Interestingly, during the earliest denoising steps, the discrepancy $\Delta_t$ is typically small in magnitude, since $x_t$ remains close to isotropic Gaussian noise. At this stage, the conditional prediction $\epsilon_\theta(x_t, c(p_+), t)$ steers the model

toward coarse, low-frequency structure, such as object placement and scene layout (Chen et al., 2025a) (e.g., if we ask for 'a cat in a box', the model starts forming 'cat-like' blobs anchored in 'box' structure). In contrast, in later steps, the attention of the denoiser is shifted to restoring the high-frequency details, and the magnitude of $\Delta_t$ turns larger. Formally, if we consider the Jacobian of the denoiser with respect to its input to see how the predicted noise updates the input:

$$J_t = \frac{\partial \epsilon_\theta(x_t, c, t)}{\partial x_t},$$ (8)

its eigen-decomposition $J_t v_i = \lambda_{i,t} v_i$ reveals the principal update directions $v_i$ in latent space, with corresponding eigenvalues $\lambda_{i,t}$ quantifying the update strength in each direction. Intuitively, each eigenvector $v_i$ defines a semantic or structural axis along which the noise prediction can perturb the latent $x_t$, while the eigenvalue determines the relative amplification or suppression along that axis. In early denoising steps, the dominant eigenvectors (those with the largest magnitude eigenvalues $|\lambda_{i,t}|$) typically align with coarse, low-frequency structure directions that correspond to high-level semantics such as object placement and global scene layout. We denote such leading directions at step $t$ as $v_{l,t}$, where $|\lambda_{l,t}| = \max(|\lambda_{i,t}|)$. The suppression effect of negative prompting along the dominant coarse-layout directions can be expressed as:

$$\text{suppression}_t^{(-)} \propto v_{l,t}^\top(-w\Delta_t) = -w\langle v_{l,t}, \Delta_t \rangle = -w||v_{l,t}|| \cdot ||\Delta_t|| \cdot \cos(\theta),$$ (9)

where $\theta$ is the angle between the coarse-layout direction $v_{l,t}$ and the suppression direction $\Delta_t$. Since $||v_{l,t}||$ is fixed by the denoiser and the prompt conditioning, and $\cos(\theta)$ is also fixed by the angle $\theta$ between the directions, the magnitude of the suppression effect is governed primarily by $||\Delta_t||$. During early steps, when $||\Delta_t||$ is small, the counterfactual prompt exerts minimal influence precisely along the directions that determine global structure. Only at later steps, when $||\Delta_t||$ grows larger, can suppression meaningfully counteract the user prompt. Thus, the dynamics of Eq. 9 explain the *lagged suppression effect*: the user condition $c(p_+)$ establishes coarse semantic anchors that shape the global layout in the early denoising steps, while the effect of the counterfactural condition $c(p_-)$ is lagged and only becomes appreciable once those structures are already formed, allowing it to attenuate but not prevent undesired effects. As a result, this vanilla negative prompting functions more as a late-stage retroactive corrector, rather than as a proactive, preventive blocker of early implausible or undesired content.

**Cumulative Trajectory Bias.** Even once $\Delta_t$ becomes substantial at later stages, the corrective capacity of the guidance remains fundamentally limited because the denoiser's predictions are always conditioned on the same latent trajectory $x_t$. As shown in Eq. 4, this trajectory has already been predominantly shaped by the original branch during the early denoising updates when updating from $x_t$ to $x_{t-1}$. Consequently, when evaluating $\epsilon_\theta(x_t, c(p_-), t)$, the input $x_t$ already encodes semantic anchors introduced by the user prompt, biasing the prediction toward those configurations. In effect, the guidance is forced to operate on latents that have inherited accumulated influence from the original prompt, attempting to correct content that is already 'locked in' by earlier conditioning. This persistent entanglement produces a *cumulative trajectory bias*, which constrains the suppressive power of the counterfactual prompt and limits its ability to fully eliminate implausible or undesired structures.

### 3.3 SYNCHRONIZED DECOUPLED GUIDANCE

To overcome the two gaps identified above, we propose **Synchronized Decoupled Guidance (SDG)**, a new guidance strategy that integrates two complementary designs. Each design is tailored to directly address one of the fundamental limitations previously identified.

**Synchronized Directional Normalization.** To mitigate the *lagged suppression effect*, we align the effects of user prompt $p_+$ and the counterfactual prompt $p_-$ from the earliest denoising steps. Rather than relying on the raw magnitude of the discrepancy $\Delta_t$, which is small when $x_t$ is still close to isotropic Gaussian noise, we focus on its direction. Specifically, we normalize the discrepancy to apply a consistent correction:

$$\hat{\epsilon}_t \leftarrow \epsilon_\theta(x_t, c(p_+), t) + \lambda \cdot \frac{\epsilon_\theta(x_t, c(p_+), t) - \epsilon_\theta(x_t, c(p_-), t)}{||\epsilon_\theta(x_t, c(p_+), t) - \epsilon_\theta(x_t, c(p_-), t)|| + \varepsilon}$$ (10)

where $\lambda$ is a scaling factor controlling the magnitude of the perturbation, and $\varepsilon$ is a small constant to ensure numerical stability. This unit-normalized directional correction emphasizes the direction of the suppression effect and makes the counterfactual prompt's suppressive influence temporally synchronized with the user prompt's constructive effect. By enforcing a direction-focused correction of constant scale, suppression remains active from the very first iteration, preventing implausible structures before they can emerge instead of only erasing them retroactively.

**Trajectory-Decoupled Denoising.** To address the *cumulative trajectory bias*, we decouple the conditioning paths of the user prompt and the counterfactual prompt. Instead of deriving both predictions from the same latent trajectory $x_t$, which has already been shaped by the user prompt and has potentially accumulated physical errors, we evolve two separate latents in parallel: one *original branch* $x_t^+$ for the user prompt $p_+$, and one *counterfactual branch* $x_t^-$ for the counterfactual prompt $p_-$. Specifically, their noise predictions are:

$$\epsilon^+ = \epsilon_\theta(x_t^+, c(p_+), t) + w \cdot (\epsilon_\theta(x_t^+, c(p_+), t) - \epsilon_\theta(x_t^+, \emptyset, t)), \tag{11}$$

$$\epsilon^- = \epsilon_\theta(x_t^-, c(p_-), t) + w \cdot (\epsilon_\theta(x_t^-, c(p_-), t) - \epsilon_\theta(x_t^-, \emptyset, t)). \tag{12}$$

By decoupling the trajectories, the counterfactual branch is free from the accumulated physical bias introduced by user-prompt conditioning. This ensures that the guidance can exert effective suppression throughout the precess, even when undesired physical phenomena would otherwise be locked into the shared trajectory.

**Summary.** By integrating both designs, SDG transforms the guidance process from a *late-stage biased retroactive corrector* into an *early-stage unbiased proactive preventer*. The final correction applied combines synchronized normalization with trajectory decoupling:

$$\hat{\epsilon}^+ = \epsilon^+ + \lambda \cdot \frac{\epsilon^+ - \epsilon^-}{||\epsilon^+ - \epsilon^-|| + \varepsilon}, \tag{13}$$

In this formulation, the user and counterfactual prompts co-evolve synchronously along distinct latent paths, and their interaction is governed by a normalized, direction-aware contrastive term. By ensuring that suppression is both non-delayed and unbiased, SDG not only overcomes the inherent limitations of negative prompting but also **maximizes the utility of our reasoning-based physical counterfactuals**. The proposed guidance strategy is able to fully empower them as proactive constraints, ensuring that implausible structures are suppressed consistently throughout the denoising process.

## 4 RESULTS

### 4.1 SETUP

**Backbones.** We evaluate our method on two representative open-source text-to-video models, *CogVideoX-5B* (Yang et al., 2025b) and *Wan2.1-14B* (Wan et al., 2025), and report results both on the base models and on their variants that are enhanced by our training-free framework.

**Compared methods.** For context, we further report: *base models*, including CogVideoX-2B (Yang et al., 2025b), LaVie (Wang et al., 2023b), VideoCrafter2 (Chen et al., 2024), Open-Sora (Zheng et al., 2024), Vchitect 2.0 (Fan et al., 2025), Cosmos-Diffusion-7B (Agarwal et al., 2025), and *physics-aware models* that incorporate additional training or bespoke modules, including PhyT2V (Xue et al., 2025), DiffPhy (Zhang et al., 2025a), VideoREPA-5B (Zhang et al., 2025b), CogVideoX-5B+WISA (Wang et al., 2025). These serve as external references to position our training-free approach.

**Benchmarks.** We evaluate on two complementary suites. *PhyGenBench* (Meng et al., 2024) provides 160 prompts spanning 27 physical laws across four domains (mechanics, optics, thermal, material) and includes an automated evaluator that reports *Physical Commonsense Alignment* (PCA). *VideoPhy* (Bansal et al., 2025) assesses real-world actions with fine-grained human-calibrated metrics for *Semantic Adherence* (SA) and *Physical Commonsense* (PC).[1]

---

[1]VideoPhy's evaluator does not have access to the user prompt at test time; it judges only the rendered video, which limits sensitivity to some fine-grained physical phenomena.

**Implementational details.** For CogVideoX-5B we use $480 \times 720$ resolution; for Wan2.1-14B, $480 \times 832$. Each video has 25 frames generated with 50 inference steps. All experiments are run on a single NVIDIA RTX 5090 (32GB).

## 4.2 COMPARISONS WITH STATE-OF-THE-ART BASELINES

| Model | Training-Free | VideoPhy SA | PC | PhyGenBench |
|---|---|---|---|---|
| *Base models* | | | | |
| CogVideoX-2B | – | – | – | 0.39 |
| LaVie | – | – | – | 0.43 |
| VideoCrafter2 | – | 0.47 | 0.36 | 0.48 |
| Open-Sora | – | 0.38 | 0.43 | 0.45 |
| Vchitect 2.0 | – | – | – | 0.45 |
| Cosmos-Diffusion-7B | – | 0.52 | 0.27 | 0.24 |
| *Physics-aware models (trained/fine-tuned)* | | | | |
| PhyT2V (Round 4) | no | 0.59 | 0.42 | 0.42 |
| DiffPhy | no | – | – | 0.54 |
| VideoREPA-5B | no | 0.72 | 0.40 | – |
| CogVideoX-5B + WISA | no | 0.67 | 0.38 | 0.43 |
| *Ours (training-free, inference-time) with two baselines* | | | | |
| CogVideoX-5B | – | 0.48 | 0.39 | 0.47 |
| CogVideoX-5B + Ours | yes | **0.49** | **0.40** | **0.49** |
| Wan2.1-14B | – | 0.49 | 0.35 | 0.40 |
| Wan2.1-14B + Ours | yes | **0.52** | **0.35** | **0.50** |

Table 1: Quantitative comparisons on VideoPhy and PhyGenBench. Our training-free SDG yields consistent gains on both backbones, with larger improvements on Wan2.1-14B and on PhyGenBench.

We first benchmark against prior base models and physics-aware systems on VideoPhy and Phy-GenBench (Tab. 1). On both CogVideoX-5B and Wan2.1-14B, adding our training-free SDG yields consistent improvements in physics-related scores; gains are modest on CogVideoX-5B and larger on Wan2.1-14B (e.g., PhyGenBench PCA $0.40 \rightarrow 0.50$). Relative to earlier base models, our SDG-enhanced variants are competitive on VideoPhy and generally stronger on PhyGenBench. We note that the VideoPhy evaluator does not access the user prompt and therefore may miss fine-grained physical cues visible only with prompt context; PhyGenBench's automated scoring can reflect such cues better. Compared with physics-aware methods that rely on additional training (e.g., PhyT2V, WISA), our approach remains competitive while requiring *no* retraining or fine-tuning, making SDG a preferred inference-time strategy.

| Model | Physical Domains (↑) | | | | |
|---|---|---|---|---|---|
| | Mechanics | Optics | Thermal | Material | Average |
| CogVideoX-5B (Baseline) | 0.43 | 0.55 | **0.42** | 0.46 | 0.47 |
| + Ours | **0.49** | **0.58** | **0.42** | **0.48** | **0.49** |
| Wan2.1-14B (Baseline) | 0.36 | 0.53 | 0.36 | 0.33 | 0.40 |
| + Ours | **0.47** | **0.60** | **0.51** | **0.40** | **0.50** |

Table 2: Quantitative comparisons of different physical domains (mechanics, optics, thermal, material). Our training-free method provides consistent gains on average and across domains.

We further analyze results across different physical domains on PhyGenBench and report its PCA in Tab. 2, comparing to both CogVideoX-5B and Wan2.1-14B baselines. Prompts are categorized into mechanics, optics, thermal, and material interactions. Our method improves performance across all four domains, showing that our method effectively generalizes and captures diverse physics phenomena. Gains are modest for CogVideoX-5B (average $0.47 \rightarrow 0.49$) but more pronounced for Wan2.1-14B (average $0.40 \rightarrow 0.50$), with particularly notable increases in thermal ($0.36 \rightarrow 0.51$) and mechanics ($0.36 \rightarrow 0.47$). These results indicate that our approach enhances the physical fidelity of diverse scenarios.

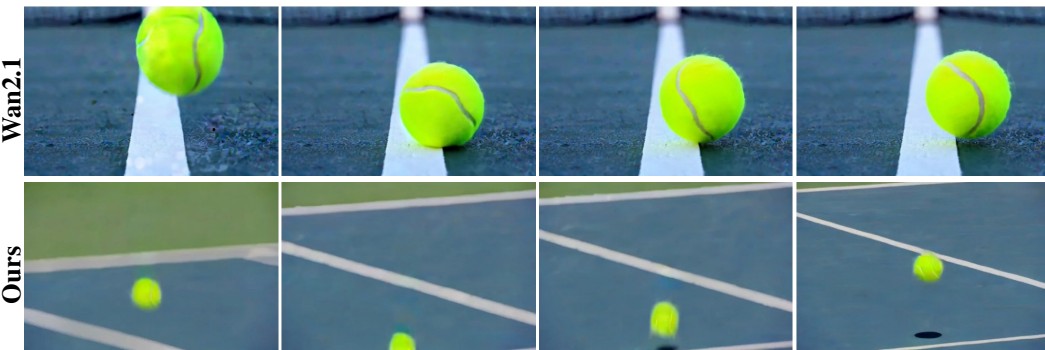

Figure 2: Qualitative comparison with Wan2.1. **Prompt**: "*A vibrant, elastic tennis ball is thrown forcefully towards the ground, capturing its dynamic interaction with the surface upon impact.*" **Baseline**: The tennis ball's motion is inconsistent with gravity-driven dynamics, with limited deformation on impact and abrupt transitions across frames. The bounce lacks elasticity. **Ours**: Our result shows a more natural downward trajectory, visible compression upon impact, and a smoother rebound trajectory, yielding a closer match to expected mechanics.

In addition to quantitative gains, we also provide qualitative comparisons with the Wan2.1 and CogVideoX baselines. As shown in Fig. 2, when simulating the dynamics of a tennis ball bouncing on the ground, Wan2.1-14B produces motion that is inconsistent with gravity-driven mechanics, exhibiting limited deformation on impact and abrupt transitions across frames. In contrast, our method generates a more natural trajectory, with visible compression upon impact and a smoother rebound, resulting in a closer match to expected elastic behavior. Similarly, Fig. 3 illustrates a scenario involving a highlighter marking cardboard. CogVideoX-5B fails to capture the proper interaction between ink and surface: strokes appear flat and disconnected from the cardboard texture, with inconsistent pen–surface contact. By comparison, our method produces strokes that adhere naturally to the surface, with ink blending seamlessly into the cardboard. These examples demonstrate improvements in both mechanics (object dynamics) and materials (object-surface interaction), reinforcing that physics-aware reasoning combined with SDG yields more physically plausible video generations. For additional qualitative comparisons across a wider set of prompts, please refer to Appendix Sec. A.1. Full video results are available in the Supplementary Material, where the dynamic effects of our approach can be more clearly observed.

## 4.3 ABLATION STUDIES

| Model | Average |
|---|---|
| Wan2.1-14B | 0.40 |
| w/o Synchronized decoupled guidance | 0.43 |
| w/o Physics-aware reasoning | 0.47 |
| w/o Synchronized directional normalization | 0.47 |
| w/o Trajectory-decoupled denoising | 0.48 |
| Full version (Ours) | 0.50 |

Table 3: Ablation experiments on PhyGenBench, reporting the average Physical Commonsense Alignment (PCA) across four domains. Removing physics-aware reasoning (PAR) reduces performance, while dropping either one of the two designs (synchronized directional normalization and trajectory-decoupled denoising) within synchronized decoupled guidance (SDG) also leads to noticeable degradation. Eliminating both designs (i.e., w/o SDG) causes an even larger drop. The full framework achieves the highest score, underscoring that PAR and both SDG designs are critical and complementary for enhancing physical plausibility.

To better understand the contributions of each component in our framework, we conduct ablation studies on PhyGenBench, as reported in Tab. 3. We examine the impact of removing the Physics-aware reasoning (PAR) module, as well as the designs within our proposed Synchronized Decoupled Guidance (SDG). SDG itself is composed of two complementary designs: Synchronized directional normalization (SDN) and Trajectory-decoupled denoising (TDD).

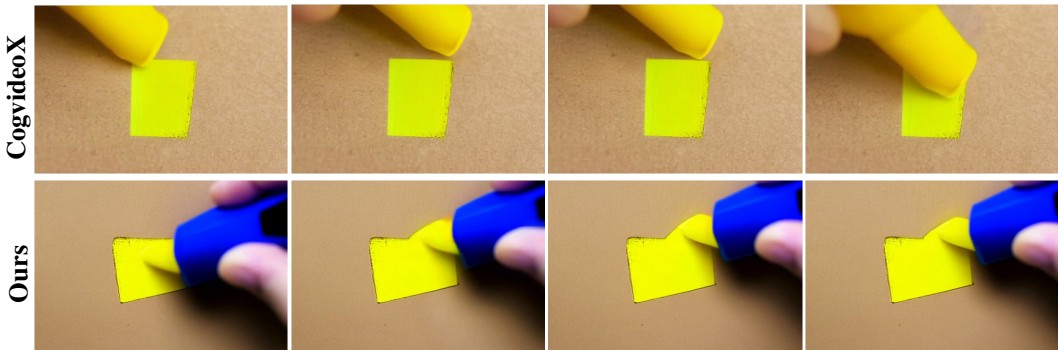

Figure 3: Qualitative comparison with CogvideoX. **Prompt**: "*A yellow highlighter is used to mark on the rough, brown surface of a cardboard, showcasing the interaction between the highlighter and the cardboard surface.*" **Baseline**: Generates inconsistent strokes, with the yellow mark appearing flat and disconnected from the cardboard's texture. The contact point with the marker is visually unconvincing. **Ours**: Produces a stroke that properly adheres to the surface, with the ink visibly blending with the cardboard texture. The pen-surface interaction is sharper and more consistent.

To enhance clarity, we provide detailed definitions of all ablation settings below:

- **w/o Synchronized Decoupled Guidance (SDG)**: Removes the entire SDG module, including both synchronized directional normalization and trajectory-decoupled denoising.
- **w/o Physics-aware Reasoning (PAR)**: Replaces our LLM-generated structured counterfactual prompts with the default negative prompt used in Wan2.1's original classifier-free guidance. This isolates the effect of the reasoning component.
- **w/o Synchronized Directional Normalization (SDN)**: Removes the first component of SDG described in Sec. 3.3, disabling the normalization and synchronization of guidance directions between the forward and counterfactual trajectories.
- **w/o Trajectory-Decoupled Denoising (TDD)**: Removes the second component of SDG described in Sec. 3.3 and reintroduces coupling between the forward and counterfactual trajectories. This version keeps inference cost identical to the full model, isolating only the effect of trajectory coupling.

The results show that the full version of our framework achieves the best overall performance, with an average score of 0.50 across the four physical domains. Removing either SDN or TDD leads to clear performance degradation (0.47–0.48 average), confirming that both designs make complementary contributions. When both are removed, i.e., in the w/o SDG variant, the performance drops further to 0.43. This demonstrates that the dual-branch design and the directional correction within SDG are both critical for enforcing consistent suppression of implausible content.

We also evaluate the effect of PAR by replacing structured reasoning with simple instructions to construct negative prompts. The w/o PAR variant achieves an average score of 0.47, which is better than the Wan2.1-14B baseline but still lower than the full version. This confirms that PAR provides more targeted and physics-aware counterfactual prompts, which empower SDG to operate effectively. A qualitative ablation study of PAR is also provided in Fig. 14, which compares counterfactual prompts generated with and without structural reasoning. Without structural reasoning, the counterfactual prompt tends to introduce irrelevant or arbitrary violations (e.g., predicting that orange juice with baking soda solidifies into a glass-like block), which are disconnected from the underlying physical process. In contrast, with structural reasoning, the LLM is guided to identify entities, interactions, and scene conditions, and then generate a counterfactual that violates the expected causal chain (e.g., the mixture remains completely still without bubbling despite the acid–base reaction). This illustrates how PAR yields higher-quality, physics-aware counterfactual prompts that directly target the intended violations of physical laws.

Overall, the ablation studies validate the importance of both major components: PAR ensures the construction of meaningful counterfactual prompts, while SDG, and specifically its two designs, SDN and TDD, ensure these prompts are fully leveraged during guidance. Together, they yield the consistent improvements observed in the full model.

## 5 CONCLUSION

We presented a training-free framework for enhancing physical plausibility in diffusion-based video generation by explicitly reasoning about implausibility and guiding the generative process. Our approach introduces a reasoning pipeline to construct counterfactual prompts that capture targeted physics-violating behaviors, and a novel *Synchronized Decoupled Guidance* (SDG) strategy that fully leverages these prompts. By addressing the two key limitations of negative prompting: lagged suppression effect and cumulative trajectory bias, through synchronized directional normalization and trajectory-decoupled denoising, SDG ensures that suppression of implausible content is both immediate and unbiased. Extensive experiments across solid mechanics, fluid dynamics, optics, and thermodynamics, along with detailed ablation studies, demonstrate that our framework significantly improves physical fidelity while preserving photorealism. This work establishes a physics-informed paradigm for video generation and highlights the potential of combining structured reasoning with inference-time guidance to advance physics-aware generative modeling.

## 6 ETHICS STATEMENT

We acknowledge that all authors of this work have read and commit to adhering to the ICLR Code of Ethics. This paper does not raise any potential violations such as harmful insights, discrimination, unfairness, privacy or security issues, or conflicts of interest. Our study does not involve human subjects, sensitive data, or applications that could cause societal harm.

## 7 REPRODUCIBILITY STATEMENT

We have made efforts to ensure the reproducibility of our work. Implementation details are provided in Sec. 4.1 and Sec. A.2. Quantitative comparisons and ablations are included in Sec. 4.2 and Sec. 4.3, and qualitative comparisons and ablations are included in Sec. A.1 and Sec. A.3 to further clarify our findings. Code will be released upon paper acceptance, and a ZIP file containing examples of generated videos is included in the Supplementary Material.

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

# A  APPENDIX

**Outline.**  This appendix provides additional results, implementation details, ablation analyses, a literature review, and a declaration on our LLM usage to further support the main paper. It is organized as follows:

- **Sec. A.1** presents *additional qualitative comparisons* with CogVideoX-5B and Wan2.1-14B across mechanics, thermodynamics, optics, and material interactions. Figures 4–9 provide side-by-side visual comparisons; Fig. 10 augments these with *automated GPT-4o assessments* from PhyGenBench of the physical plausibility of each video; and Fig. 11 further compares our approach against *negative prompting (NP) within CFG*, highlighting the limited gains of NP relative to our SDG. Together, these examples complement the quantitative results by showing improved fluid-object interactions, material transformations, and object dynamics.

- **Sec. A.2** provides *additional implementation details* for reproducibility. Fig. 12 shares the *instruction template* used to guide the LLM, including constraints and strict output format; Fig. 13 illustrates *worked examples* across domains (optics and thermodynamics), demonstrating how the analysis stage grounds the subsequent counterfactual.

- **Sec. A.3** reports a *qualitative ablation of Physics-aware Reasoning (PAR)* for counterfactual prompt construction (Fig. 14). We compare counterfactuals generated *with vs. without* structured reasoning for thermodynamics prompts and show that PAR yields targeted, physics-aware violations (e.g., condensation) rather than generic negatives.

- **Sec. A.4** provides a *literature review* that summarizes related works and highlights the gaps our method addresses.

- **Sec. A.5** provides a *declaration* on our LLM usage.

All figures include detailed captions to support discussion and analysis of the findings. For completeness, the *Supplementary Material* additionally contains full videos of all qualitative examples, where the physical dynamics are best appreciated in motion.

## A.1  ADDITIONAL QUALITATIVE COMPARISONS

This section provides additional qualitative comparisons between our method and the CogVideoX-5B and Wan2.1-14B baselines across prompts spanning mechanics, thermodynamics, optics, and material interactions.

Figures 4-9 present side-by-side comparisons, where baseline models often generate visually appealing sequences but overlook key physical processes, such as the absence of condensation during boiling, incomplete material phase transitions, or unrealistic object-surface interactions. In contrast, our method produces outcomes that better align with physical commonsense: for example, more coherent fluid-object interactions (Fig. 4, 5), smoother material transformations (Fig. 8, 9), and more faithful object dynamics (Fig. 2, 3).

Beyond visual inspection, Fig. 10 shows qualitative results accompanied by evaluations generated by PhyGenBench's automatic evaluator through the GPT-4o API, which assess the physical plausibility of each video. Finally, Fig. 11 compares our approach not only with the baselines but also with negative prompting (NP) within classifier-free guidance, highlighting that NP yields only limited improvements while our Synchronized Decoupled Guidance (SDG) effectively mitigates the shortcomings. These qualitative studies complement our quantitative evaluations and illustrate how combining Physics-aware Reasoning (PAR) with SDG improves physical plausibility while preserving photorealism, all without retraining or fine-tuning.

We provide detailed per-example captions in this section, and please refer to the Supplementary Material for full video results, where the dynamics can be best appreciated.

## A.2  ADDITIONAL IMPLEMENTATION DETAILS

We include further implementation details to improve reproducibility of our physics-aware reasoning pipeline. Fig. 12 provides the instruction template used to guide the LLM during counterfactual

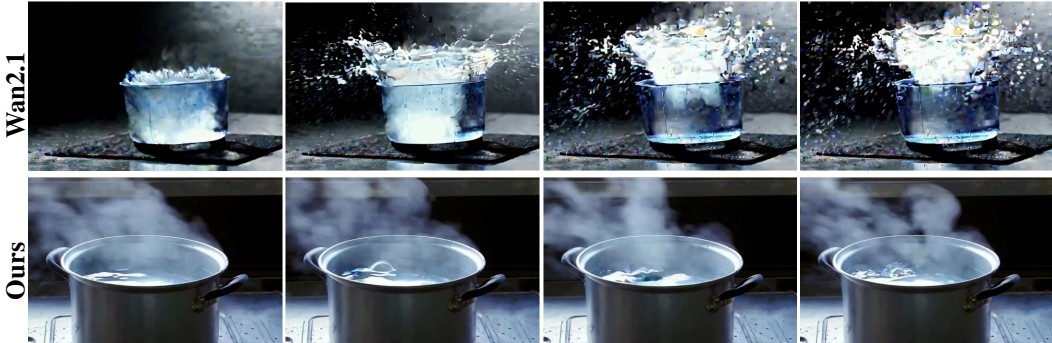

Figure 4: Qualitative comparison with Wan2.1. **Prompt**: "*A silver spoon is slowly inserted into a glass of crystal-clear water, revealing the fascinating visual changes and reflections as the spoon interacts with the liquid.*" **Baseline**: The generated sequence struggles to capture realistic refraction and liquid interaction. The spoon appears disconnected from the water surface, and the reflections lack physical plausibility. **Ours**: Our method produces a coherent depiction of the spoon entering the water, with realistic ripples, refraction, and surface reflections. This creates a more physically faithful impression of object-fluid interaction.

Figure 5: Qualitative comparison with Wan2.1. **Prompt**: "*A timelapse captures the transformation of water in a pot as the temperature rapidly rises above 100°C.*" **Baseline**: The sequence unrealistically depicts explosive splashes, ignoring the gradual bubbling and vapor release expected from water heating above 100°C. **Ours**: Our method captures progressive bubbling and the formation of rising vapor clouds, consistent with the condensation process. This produces a more physically plausible thermal interaction.

prompt construction. The template specifies that the LLM should first output a structured analysis describing entities, environments, interactions, and temporal evolution of the event, followed by a counterfactual description that is visually plausible yet physically implausible. It also enforces key requirements such as maintaining subjects and settings, avoiding repetition, and ensuring clear violations of physical laws, and it defines a strict output format to ensure consistency. Fig. 13 further illustrates two representative examples of physics-aware reasoning across different domains. In the optics case, the model analyzes refraction through a magnifying glass and generates a counterfactual where the embossing shrinks instead of enlarging. In the thermodynamics case, the model reasons about heat transfer and the phase transition of butter, then generates a counterfactual where butter is fully liquefied from the start without any melting process. These examples highlight how the LLM is able to identify relevant entities, interactions, and governing principles, and then construct counterfactuals that are both plausible to the viewer and explicitly violate physical laws. Lastly, for the guidance strength of SDG in Eq. 13, we find $\lambda = 30$ to be the best choice in general.

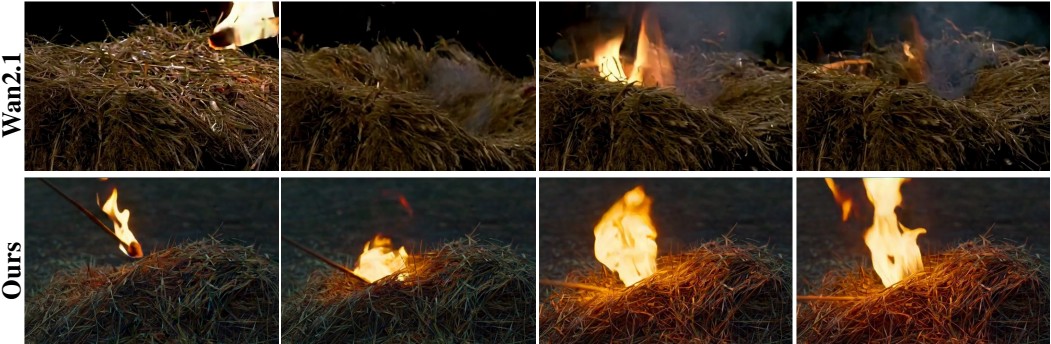

Figure 6: Qualitative comparison with Wan2.1. **Prompt**: "*A small burning stick was thrown into a pile of hay.*" **Baseline**: The ignition of hay is abrupt and spatially inconsistent, with flames appearing unnaturally large and sudden. **Ours**: Our model shows fire propagating gradually from the burning stick to the hay, with smoother flame development and more realistic local ignition dynamics.

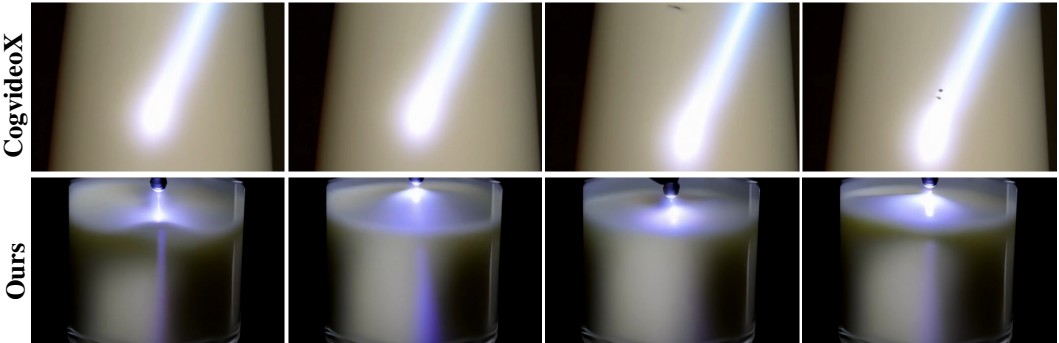

Figure 7: Qualitative comparison with CogvideoX. **Prompt**: "*A concentrated, bright beam of light generated by a laser pointer is passing through a glass of thick whole milk, creating a mesmerizing display as the light interacts with the milk's particles, casting intricate patterns and subtle hues within the fluid.*" **Baseline**: The light beam appears static and detached from the milk medium, with minimal scattering or hue variation, failing to show how light interacts with particles in the liquid. **Ours**: Our sequence captures a concentrated beam penetrating the milk, producing scattering and subtle glow effects that vary realistically across frames, aligning with optical refraction principles.

## A.3 QUALITATIVE ABLATION OF PHYSICS-AWARE REASONING

Figure 14 shows a qualitative ablation of physics-aware reasoning (PAR) for counterfactual prompt construction. We present two thermodynamics-related prompts with highly similar descriptions. Without PAR, the generated counterfactual prompts are overly generic and lack specificity, failing to capture the relevant physical phenomenon (e.g., condensation). In contrast, with PAR, the LLM first infers the detailed underlying process and then produces counterfactuals that are not only more physically grounded but also visually realistic. This demonstrates that structured reasoning is essential for generating counterfactual prompts that directly target meaningful violations of physical laws.

## A.4 RELATED WORK

**Video generative models.** Video generative modeling has rapidly progressed by extending image generative frameworks to capture temporal dynamics (Blattmann et al., 2023; Wang et al., 2023a; Chen et al., 2024; Girdhar et al., 2023). Early diffusion-based approaches, such as Video Diffusion Models (Ho et al., 2022b), adopted 3D convolutional architectures to extend denoising diffusion probabilistic models (DDPMs) (Ho et al., 2020; Nichol & Dhariwal, 2021) into the video domain, but were limited in scale and realism. Subsequent advances leveraged pretrained text-to-image

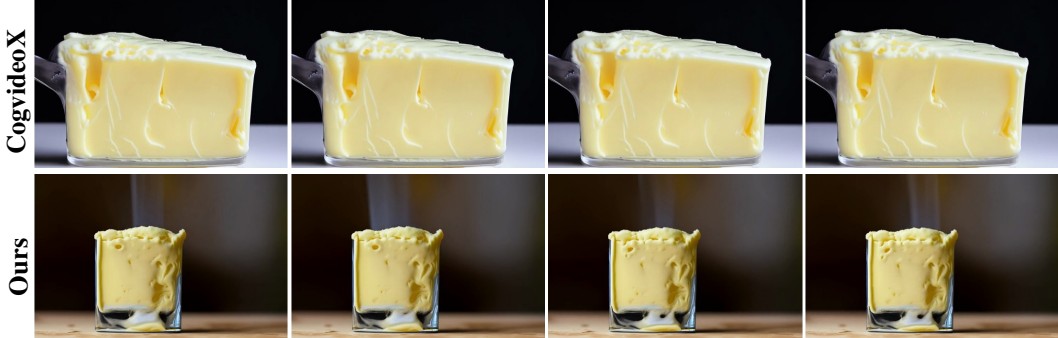

Figure 8: **Prompt**: "*Qualitative comparison with CogvideoX. A timelapse captures the gradual transformation of butter as the temperature rises significantly.*" **Baseline**: The butter remains largely unchanged, with rigid textures and little indication of gradual phase transition. The thermal process is not conveyed. **Ours**: Our method depicts butter softening and progressively melting, accompanied by rising vapor. This better reflects the heat-driven transition from solid to liquid.

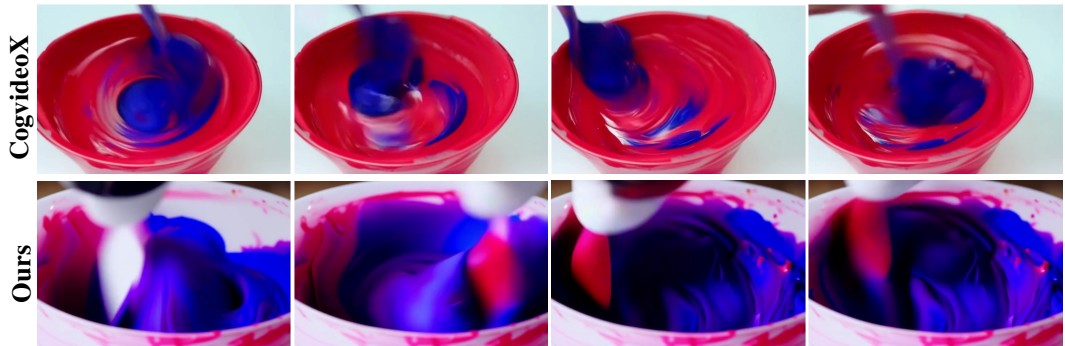

Figure 9: Qualitative comparison with CogvideoX. **Prompt**: "*Equal amounts of red and blue paint are rapidly combined, with the mixture being vigorously stirred until fully blended.*" **Baseline**: The mixing of red and blue paint is incomplete and static, with colors remaining largely separated. The blending dynamics are underdeveloped. **Ours**: Our sequence shows vigorous stirring, with swirling patterns and gradual blending into purple, consistent with fluid mixing behavior.

(T2I) models, notably Stable Diffusion (Rombach et al., 2022), to build stronger text-to-video (T2V) systems. Make-A-Video (Singer et al., 2022) and Imagen Video (Ho et al., 2022a) pioneered this paradigm, showing that reusing large T2I backbones and augmenting them with temporal layers could produce plausible short clips. Other systems such as Runway Gen-1 (Esser et al., 2023) extended controllability by incorporating text, image, and video conditions for editing and stylization.

The field has since advanced through architectural innovations and scaling. Diffusion Transformers (DiTs) (Peebles & Xie, 2022) demonstrated strong spatiotemporal modeling capacity, enabling models such as Open-Sora (Zheng et al., 2024), Cosmos (Agarwal et al., 2025), CogVideoX (Yang et al., 2025b), HunyuanVideo (Kong et al., 2024), Kling (Kuaishou, 2024), and Wan2.1 (Wan et al., 2025) to achieve substantial gains in video quality, motion realism, and scalability. Beyond raw scale, several works target stronger spatiotemporal structure and control: Step-Video-T2V (Ma et al., 2025) couples a deep-compression Video-VAE with a 30B DiT trained via flow matching to extend clip length and bilingual prompting; GEN3C (Ren et al., 2025) introduces 3D-informed, camera-consistent generation; and Tora (Zhang et al., 2025c) studies trajectory-oriented DiT design for longer, coherent motion. Proprietary systems such as Sora (OpenAI, 2024), Gen-3 (Runway, 2024), and Google DeepMind's Veo series (Google DeepMind, 2025) have further captured public attention by producing long, high-fidelity videos with rich dynamics. These milestones collectively

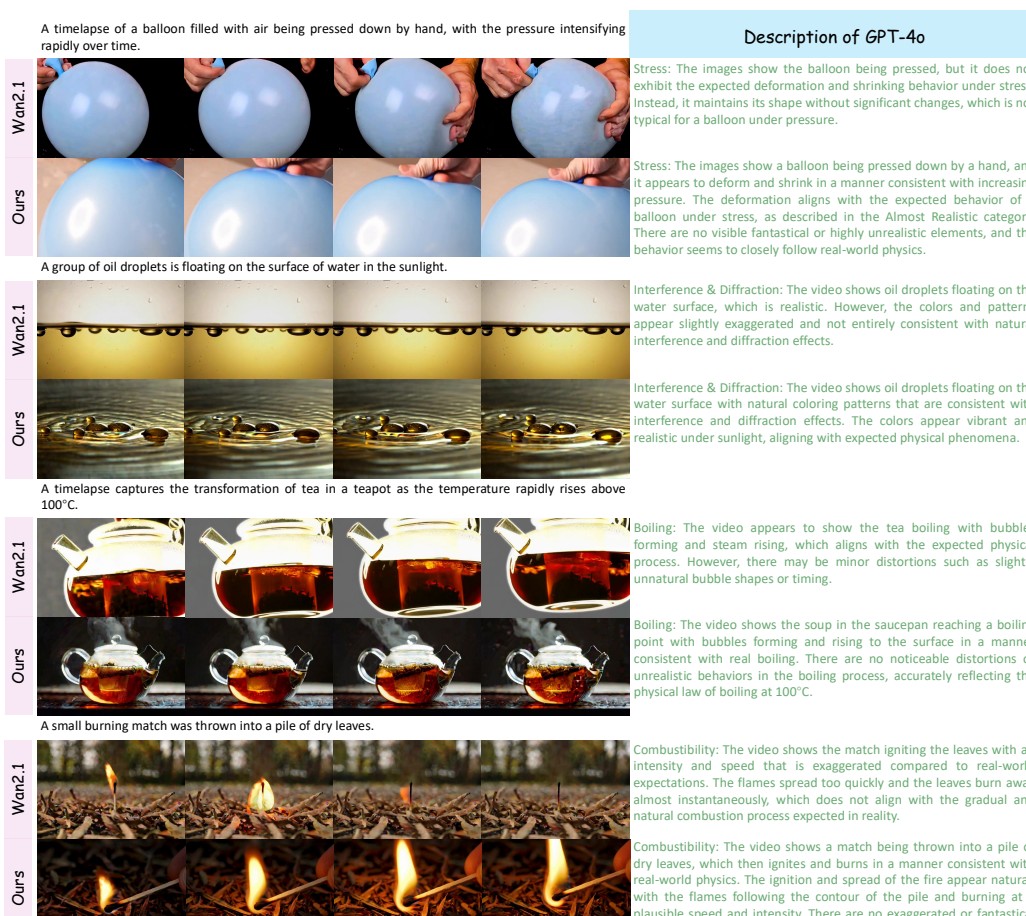

Figure 10: Additional qualitative samples generated by our model across diverse prompts. Alongside the visual results, we include the evaluations provided by GPT-4o, invoked through the automatic evaluator of PhyGenBench, which assesses the overall physical plausibility of each video. Please refer to the Supplementary Material for full video results, as the physical dynamics are best appreciated in motion.

underscore the effectiveness of scaling DiT-based architectures and leveraging massive video-text datasets.

Despite these successes, existing video generative models primarily fit data distributions drawn from large-scale internet corpora, where explicit representations of physical laws are rare and physical phenomena are underrepresented. As a result, even state-of-the-art systems often produce videos that deviate from physical commonsense, for instance, fluids ignoring gravity or phase transitions behaving unrealistically. Our work is motivated by this gap: while recent T2V models have achieved remarkable photorealism and temporal consistency, ensuring compliance with real-world physics remains an open challenge.

**Physics-aware video generations.** Researchers have increasingly focused on improving and evaluating the physical consistency of generated videos (Liu et al., 2025; Motamed et al., 2025; Lin et al., 2025; Zhao et al., 2025; Li et al., 2025; Chen et al., 2025b;c; Yang et al., 2025a; Wong et al., 2025; Xie et al., 2025). One line of effort has been to build dedicated benchmarks. For example, VideoPhy (Bansal et al., 2025) evaluates real-world actions using fine-grained human judgments across semantic adherence, physical commonsense, and explicit rule violations. PhyGenBench (Meng et al., 2024) curates 160 prompts spanning 27 physical laws across four domains and introduces an automated evaluator for physical commonsense alignment. Together, these resources have revealed that

even state-of-the-art video diffusion models frequently generate outputs that deviate from real-world physics.

In parallel, several works attempt to explicitly encode physical constraints into generative processes. Early approaches such as DANO (Cleac'h et al., 2022), MotionCraft (Aira et al., 2024), and Phys-Gen (Liu et al., 2024) parse objects from static images and estimate their rigid-body dynamics in a differentiable manner, then animate these estimates into short videos. While interpretable, these pipelines are limited to predefined physical categories (rigid motion) and static scenarios, which hinders their applicability to complex or diverse phenomena.

More recent models have pursued broader physics-awareness within diffusion-based video generation. PhyT2V (Xue et al., 2025) uses large language and vision-language models to detect inconsistencies in generated videos and iteratively refine prompts with physics-based feedback, though this introduces substantial inference overhead. Then, several contemporary works also contribute to this effort. For example, DiffPhy (Zhang et al., 2025a) integrates differentiable physics simulation into the training loop, encouraging the generator to respect Newtonian laws, but requires re-training on curated physics datasets. VideoREPA (Zhang et al., 2025b) incorporates structured physical signals during pre-training to enhance physical perception, while WISA (Wang et al., 2025) augments training data with explicitly annotated physical phenomena, enabling the model to learn structured physical priors. Despite their promising results, all these methods rely on additional training or fine-tuning.

By contrast, our framework is training-free and inference-time only. We introduce physics-aware reasoning (PAR) to construct targeted counterfactual prompts that deliberately violate governing laws, and Synchronized Decoupled Guidance (SDG) to suppress implausible generations. This allows us to improve physical plausibility on strong backbones without the cost of retraining, offering a complementary direction to recent physics-aware efforts.

## A.5 DECLARATION ON LLM USAGE

LLM is only used to aid or polish writing in addition to facilitating the physics-aware reasoning for constructing the counterfactuals (Sec. 3.1). We have also provided *additional implementational details* in Sec.A.2, including the *instruction template* used to guide the LLM, including constraints and strict output format (Fig. 12), and some *worked examples* of leveraging LLM for constructing counterfactual prompts (Fig. 13).

**(+):** A metal fork is gently placed into a glass of crystal-clear water, displaying the interesting visual distortions and reflections as the fork meets the liquid.
**(-):** A metal fork is gently placed into a glass of crystal-clear water, but the entire utensil appears perfectly straight and continuous through the surface, with no visible refraction or distortion.

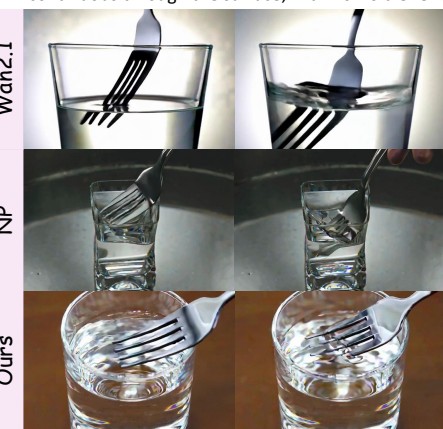

**Wan2.1**

The images show the fork with **exaggerated distortions and large ripples** that are not consistent with the subtle effects of refraction. The visual artifacts are overly pronounced, indicating a detachment from the expected physical properties of light and water interaction.

**NP**

The video shows the metal fork with **subtle distortions** at the point where it meets the water. The distortions are not overly exaggerated, but there are minor deviations from expected refraction angles, indicating slight inaccuracies in the visual representation of refraction.

**Ours**

At the water surface, the fork appears slightly misaligned between the part above and the part submerged in water. This small offset is **consistent with light bending at the air–water interface**. The effect is realistic and visually coherent.

**(+):** A timelapse captures the transformation of soup in a saucepan as the temperature rapidly rises above 100°C.
**(-):** A timelapse captures the transformation of soup in a saucepan as the temperature rapidly rises above 100 °C, yet the soup remains completely still and silent, with no bubbling or surface disturbance throughout the heating process.

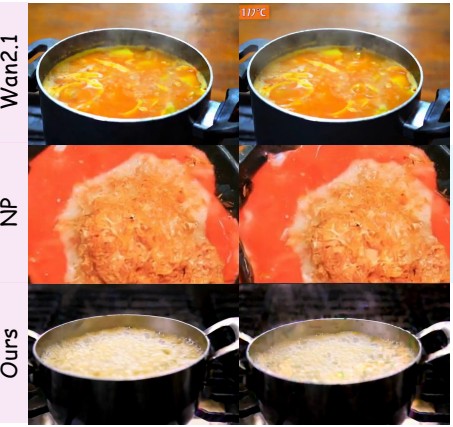

**Wan2.1**

The video shows the soup reaching a temperature of 117°C, which is above the normal boiling point of water (100°C) at standard atmospheric pressure. Despite this, the soup **does not exhibit the expected vigorous boiling** with a large number of bubbles surging to the surface.

**NP**

The video shows a scene where the soup **does not appear to boil** despite reaching a temperature above 100°C, which contradicts the physical law that states the soup should boil and bubbles should rise to the surface.

**Ours**

The video shows the soup in the saucepan reaching a boiling point with bubbles forming and rising to the surface in a manner consistent with real boiling. There are no noticeable distortions or unrealistic behaviors in the boiling process, **accurately reflecting the physical law of boiling at 100°C**.

**(+):** A vibrant, elastic tennis ball is thrown forcefully towards the ground, capturing its dynamic interaction with the surface upon impact.
**(-):** A vibrant, elastic tennis ball is thrown forcefully towards the ground, but instead of bouncing, it collapses and clings flat against the surface, refusing to rebound despite its elastic structure.

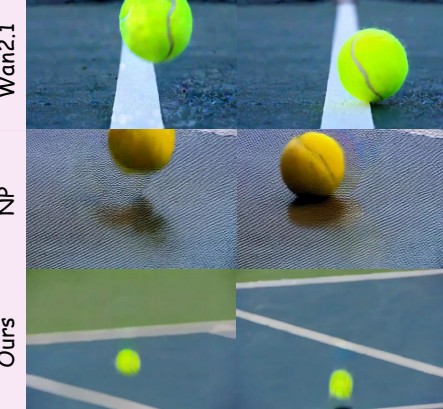

**Wan2.1**

The images show the tennis ball hitting the ground and then **rolling without bouncing back up**, which contradicts the expected behavior of an elastic ball. This behavior disregards the laws of elasticity, as the ball should bounce back up upon impact.

**NP**

The yellow ball touches and **rolls/spreads on the wavy surface without showing visible compression or a rebound trajectory**. The video does not exhibit the expected elastic bounce behavior; it shows more of a rolling or sliding motion instead.

**Ours**

The video shows a tennis ball hitting the ground **and bouncing back up in a manner consistent with the principles of elasticity**. There are no noticeable distortions or deviations from expected behavior, such as unrealistic changes in speed, height, or angle. The interaction between the ball and the surface appears to align well with real-world physics

Figure 11: Additional comparisons with both the baseline and using negative prompting (NP) in CFG. The symbols (-) and (+) denote user prompts and counterfactual prompts, respectively. The descriptions on the right report the overall physical plausibility of the generated videos, as assessed by PhyGenBench's automatic evaluator through the GPT-4o API. As highlighted in orange, our method effectively mitigates the shortcomings of the baseline, whereas NP yields only limited improvement.

## Instruction for Physics-Aware Reasoning

You are a physicist specializing in **<main physical category / states of matter>**. Your task is to construct captions for a video generation system that reflect physically accurate processes involving **<main physical category / states of matter>**. Your goal is to highlight physical phenomena by producing both physically accurate descriptions and counterfactual variants that violate the underlying physical laws.

*For each input caption tag describing a physically valid event:*
1. Generate a precise phenomenological *Analysis* that includes:
   (i) Identification of the primary **physical entities and environment**.
   (ii) A temporal description of the process, from initiation to conclusion.
   (iii) Key **physical interactions** governed by the stated physics category.
2. Create a *Counterfactual Caption* that describes a visually plausible but physically impossible version of the same phenomenon. This counterfactual must adhere to the following constraints:

*Counterfactual Caption Requirements:*
  (i) The **subjects and setting** must be the same as those in the original caption.
  (ii) The caption should describe a **dynamic process** suitable for video generation, avoiding purely static or abstract scenarios.
  (iii) The event described must be **physically implausible under the given conditions**, while remaining **visually realistic and compelling**.
  (iv) The counterfactual scenario must not describe a phenomenon that could plausibly occur in any slightly modified or analogous setting, ensuring that viewers recognize the violation through visual cues alone, without needing the caption.
  (v) Each violating caption must be **unique** across the dataset and **non-repetitive**.
  (vi) The counterfactual event must be **simple and focused**: it may include elements that exist in reality, but their behavior must violate the relevant physical laws within the specific context.
  (vii) Avoid absurd or comically exaggerated scenarios. The violation should feel subtle and believable at first glance, but clearly inconsistent with physical principles upon scrutiny.

*Example (good counterfactual):*
"In a space station, a hand tilts a cup of water. As the water exits the cup, it freezes instantly mid-air, transforming into a cluster of rigid, perfectly hexagonal ice crystals that remain suspended and motionless, maintaining their geometric shape without melting or drifting."

*Output Format (strict):*
All outputs must be returned in a Python dictionary structure with the following keys:
{
  'Analysis': '<Detailed physical description>',
  'Counterfactual Caption': '<Single-sentence, visually plausible but physically impossible description>'
}

Figure 12: Instruction template used for physics-aware reasoning. The LLM is prompted to generate counterfactual captions by first producing a structured analysis that identifies entities, environments, interactions, and temporal evolution, followed by a counterfactual description that is visually plausible yet physically implausible. The figure specifies key requirements for constructing counterfactuals (e.g., maintaining subjects and settings, ensuring non-repetition, and enforcing clear violations of physical laws), provides an example of a valid counterfactual, and defines the strict output format. These implementational details are included to enhance reproducibility and to ensure consistent generation of physics-aware counterfactual prompts across experiments.

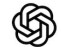 User      model: GPT-4o

**e.g. Category of Optics**

**Instruction:** You are a physicist specializing in optics …

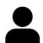 **Caption:** A magnifying glass is gradually moving closer to a coin, revealing the intricate details and textures of the embossed design as it approaches.

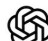 {
 **'Analysis':** 'The primary optical entities in this scene are the magnifying glass, the coin, and the ambient light source illuminating the scene. The environment is brightly lit, likely indoors, allowing for clear observation of the coin's details. As the magnifying glass moves closer to the coin, the light passing through the lens undergoes refraction due to the curved glass surface. This refraction bends the light rays, magnifying the image of the coin beneath it. Over time, as the distance between the magnifying glass and the coin decreases, the level of magnification increases, making the intricate textured details of the coin's embossed design more visible. The key optical principle involved here is refraction, governed by Snell's Law, which explains how light bends when it passes through different media with varying refractive indices.',
 **'Counterfactual Caption':** 'As the magnifying glass moves closer to the coin, the embossing details appear to shrink rather than enlarge, with the design becoming less visible and more blurred.'
}

**e.g. Category of Thermal**

**Instruction:** You are a physicist specializing in thermodynamics …

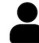 **Caption:** A timelapse captures the gradual transformation of butter as the temperature rises significantly.

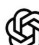 {
 **'Analysis':** 'The process involves butter, which is initially solid, placed in an environment where the temperature is gradually increased. As the temperature rises, the butter begins to soften and eventually reaches a point where it starts to melt, transitioning from a solid to a liquid state. This transformation occurs over time, as the heat energy is absorbed by the butter, leading to a change in its molecular structure. The governing principles include the phase transition dynamics of fats and the conservation of energy, where heat is transferred from the environment into the butter, causing it to melt progressively.',
 **'Counterfactual Caption':** 'The timelapse immediately shows fully liquefied butter from the start, with no visible temperature increase or melting process observed.'
}

Figure 13: Examples of physics-aware reasoning for counterfactual prompt construction across different physical domains. In the optics case (top), the model analyzes how light refracts through a magnifying glass and generates a counterfactual where the embossing shrinks rather than enlarges. In the thermodynamics case (bottom), the model reasons about heat transfer and phase transition of butter, then generates a counterfactual where butter is fully liquefied from the start, with no observable melting process. These examples illustrate how physics-aware reasoning allows the LLM to identify relevant entities, interactions, and governing principles, and then produce counterfactuals that are both visually plausible and explicitly violate the expected physical laws.

**Ablation of Physics-Aware Reasoning**

**Original Prompt**

① A timelapse captures the transformation as water vapor in a humid environment comes into contact with a cool glass surface.
② A timelapse captures the transformation as steam in a kitchen environment comes into contact with a cool metal pot lid.

**w/o Physics-Aware Reasoning**

① ***Counterfactual Caption:*** 'A timelapse captures the transformation as water vapor in a humid environment instantly freezes into solid ice upon contact with a cool glass surface. '
② ***Counterfactual Caption:*** 'A timelapse captures the transformation as steam in a kitchen environment comes into contact with a cool metal pot lid, instantly freezing into a solid layer of ice. '

**w/ Physics-Aware Reasoning**

① {***Analysis:*** 'The process involves water vapor in the air coming into contact with a cooler glass surface, causing the vapor to condense into liquid water droplets due to the temperature difference. This transformation is governed by the principles of condensation, where the vapor reaches its dew point upon touching the cooler surface, releasing latent heat and forming droplets. Over time, these droplets may grow larger as more vapor condenses, eventually coalescing and possibly dripping down the surface. This process is influenced by the ambient humidity, the temperature of the glass, and the surrounding air pressure. '
    ***Counterfactual Caption:*** 'The glass surface is instantly covered in water droplets from the beginning, without any observable condensation or gradual droplet formation. '}
② {***Analysis:*** 'The process begins with steam, which is water vapor in a gaseous state, rising in a kitchen environment where a pot is in use. This steam comes into contact with a cooler metal pot lid. As the steam touches the lid, it loses heat to the cooler surface, causing it to condense into liquid water droplets. This transformation from gas to liquid occurs gradually, as the heat transfer continues, and more steam condenses over time. The governing principles include heat transfer, specifically thermal conduction from the steam to the metal, and phase transition dynamics as the steam reaches its dew point and condenses into water droplets. ',
    ***Counterfactual Caption:*** 'The pot lid is already covered in droplets from the start, with no observable steam or condensation process taking place. '}

Figure 14: Qualitative ablation of physics-aware reasoning for counterfactual prompt construction. We show two thermodynamics-related prompts with similar descriptions. Without physics-aware reasoning, the generated counterfactual prompts are generic and lack specificity, failing to capture the relevant physical process (e.g., condensation). In contrast, with physics-aware reasoning, the LLM first infers the detailed underlying physical phenomenon and then produces counterfactuals that are both more physically grounded and visually realistic.

