# OpenReview forum: "Enhancing Physical Plausibility in Video Generation by Reasoning the Implausibility"
_ICLR.cc/2026/Conference — Submitted to ICLR 2026_

### Official Review · Reviewer_TVE5 · 2025-10-23

**Soundness:** 2
**Presentation:** 3
**Contribution:** 2
**Rating:** 4
**Confidence:** 3

**Summary:**

This presents a novel training-free framework aimed at increasing the physical realism of videos produced by diffusion-based text-to-video models. The authors observe that while such models can generate visually compelling content, they often produce motions and interactions that defy intuitive physical laws—for example, objects moving without apparent forces or behaving in ways inconsistent with gravity.

To address these issues, the paper proposes a two-part method. First, a Physics-Aware Reasoning (PAR) pipeline automatically constructs “counterfactual prompts” that encode physically implausible scenarios: given a user prompt, PAR identifies entities, interactions and context, then generates an alternative prompt that preserves the scene yet violates some physical law. Second, the paper introduces Synchronized Decoupled Guidance (SDG), a guidance strategy applied during inference that leverages these counterfactual prompts to steer the generation away from implausible content. SDG has two key mechanisms: Synchronized Directional Normalization (SDN), which ensures early suppression of undesired directions, and Trajectory-Decoupled Denoising (TDD), which prevents the negative prompt’s suppression being weakened by trajectory bias.

Empirical results across multiple physical-domains (mechanics, optics, thermal, material) show that this approach improves metrics of physical-fidelity while preserving photorealism, and does so without retraining the base diffusion model. Ablation studies further indicate that both PAR and each component of SDG contribute meaningfully to the gains.

**Strengths:**

1. The authors provide a clear articulation of their motivation (video generation models frequently violate intuitive physical laws), a clean decomposition of the solution into two parts (PAR and SDG), and description of the mechanisms involved. The framing is intuitive (reason about “what’s impossible”, then guide to avoid it). The paper includes explicit discussion of the failure modes they address (lagged suppression, trajectory bias) which helps the reader understand why previous simple negative prompting may fail. The clarity of presentations, at least from the abstract and summary, is good.
2. The paper proposes a fresh and unconventional viewpoint: instead of only teaching a video-generation model to learn physical laws, it teaches the model to avoid physically implausible scenarios by reasoning about what should not happen. This “counterfactual prompt + inference-time guidance” paradigm is a creative combination of negative prompting, physical reasoning, and guidance design. The use of a “physics-aware reasoning” pipeline that constructs specifically tailored counterfactual prompts is particularly novel. Moreover, the subsequent guidance approach, dubbed “Synchronized Decoupled Guidance”, introduces non-standard mechanisms (synchronized directional normalization, trajectory-decoupled denoising) that address shortcomings of naive negative prompting.
3. The significance of the work is high, for several reasons. First, physical plausibility is a major challenge for generative video models: as visual fidelity improves, the next frontier is coherence, causality, and physical realism. A plug-and-play, training-free method that improves physical plausibility is of wide interest. Second, since the method does not require retraining or redesigning the base model, it is more likely to be adopted in practice, thus raising its impact. Third, by reframing the problem (reasoning about implausible content), the work may inspire future research in generative modelling, negative-prompt design, and physical reasoning. Finally, the fact that the approach crosses multiple physical domains suggests broader applicability, rather than being narrowly tailored.

**Weaknesses:**

1. Although the authors present quantitative gains, the magnitude of improvement appears quite small in certain cases. For example, in Table 1 the enhancement reported on the VideoPhy dataset for the model CogVideoX‑5B is minimal which suggests very limited practical impact. Furthermore, the qualitative examples (e.g., Figures 2 and 3) do not clearly demonstrate visually striking improvements. Even in the supplementary material—a group of oil droplets floating on water is claimed to be better—one can observe the droplets behaving as if “stuck” on the surface, which is itself inconsistent with correct fluid behavior. This suggests that while the framework aims to boost physical plausibility, the perceptual gains may not yet be compelling or unambiguous.
2. The experimental results show inconsistent benefits across backbone models: while the method improves performance more significantly for Wan2.1‑14B, the gains on CogVideoX-5B are much smaller (0.47 to 0.49). This raises concerns about whether the proposed inference-time framework generalises broadly across different text-to-video generation models. The authors might have strengthened their claims by including a wider range of baselines (different architectures, training conditions) or more challenging datasets. Without that breadth, it remains unclear how universally applicable the method is and whether it is robust in diverse settings.
3. The authors note in lines 162–168 that user prompts often lack explicit specification of physical processes, and therefore rely on prompt engineering via their pipeline to supply the “missing” physical context. This reliance on engineered prompt augmentation may undermine the claim that the underlying video generation model is being endowed with stronger physical commonsense or reasoning capabilities. Instead, the results might simply reflect improved prompt-following behaviour rather than enhanced physical modelling per se. In other words, a general-purpose world simulator should internally represent and reason about physical laws rather than being steered via language-based prompt tweaks. As such, while the method may produce somewhat more plausible videos, it remains unclear how much the model’s intrinsic physical understanding has been improved, versus how much the quality stems from smarter prompting.

**Questions:**

I don't see the implementation in the supplementary material. Will this be opensourced?

---

> ### Author Response · Authors · 2025-11-26
>
> Thank you for the reviewer’s constructive and insightful comments. Below we address each concern in detail.
>
> # Response to W1
>
> **(1) On the improvement observed on VideoPhy**
>
>
> As clarified in our paper (page 6, footnote), the VideoPhy evaluator does not have access to the user prompt during testing. It assesses only the generated video, without knowledge of the underlying objects, materials, or expected physical state transitions. Consequently, the Physical Commonsense (PC) metric is inherently less sensitive to fine-grained physical improvements that rely on interpreting the prompt’s physical setup.
>
> **(2) On qualitative results**
>
> **In Figure 2**, the baseline Wan2.1 produces a tennis ball that hits the ground and then slides with small unnatural oscillations, failing to exhibit proper elastic rebound. In contrast, our SDG-enhanced result demonstrates:
> - **a realistic downward trajectory,**
> - **visible compression on impact,**
> - **a physically consistent rebound motion.**
>
>
> **In Figure 3**, the baseline video shows significant physical inconsistencies in the interaction between the highlighter and the cardboard surface. The color appears before the pen even touches the surface, and the pigment does not align spatially with the pen tip. Our method eliminates these issues:
>
> - **ink appears exactly at the contact point and with correct timing,**
> - **producing a coherent material-interaction process.**
>
>
> **In the “oil droplets on water” example** from the supplementary material, the scenario falls under the optics category of PhyGenBench Benchmark, where the evaluation primarily targets interference and diffraction effects rather than fluid-mechanical behavior. The official auto evaluator explicitly judges our output as more physically plausible in the optical sense:
>
> - **“Our video shows oil droplets floating on the water surface with natural coloring patterns that are consistent with interference and diffraction effects. The colors appear vibrant and realistic under sunlight, aligning with expected physical phenomena.”**
>
> This indicates that the improvement lies in reproducing optical effects, which is the intended physical domain of the prompt. The “stuck” behavior noted by the reviewer relates instead to mechanics (e.g., interfacial forces or buoyancy), which is outside the scope of this particular benchmark category. Extending our method to simultaneously optimize plausibility across multiple physics domains is a direction we are actively exploring.
>
> # Response to W2
>
> To further assess the generalization ability of our framework across different backbone architectures, we additionally evaluate our method on Vchitect-2.0 (experiment settings: number of frames = 40, fps = 8, guidance scale = 7.5). The results are shown below:
>
> **Comparison with Vchitect-2.0 Baseline**
>
> | **Model**             | **PhyGenBench** |
> |-----------------------|:---------------:|
> | Vchitect-2.0          | 0.46            |
> | **Vchitect-2.0 + Ours** | **0.49**        |
>
> The enhancement on a third, architecturally distinct backbone provides direct evidence that our method is not tied to a particular model family and generalizes well across diverse text-to-video generators.

---

> > ### Author Response · Authors · 2025-11-26
> >
> > # Response to W3
> >
> > **(1) Isolating Prompt Engineering Effects Through Ablation**
> >
> > In the ablation study (Table 3), we included the w/o Synchronized Decoupled Guidance variant to isolate the effect of prompt engineering alone. In this variant, we replace the conventional negative prompt in CFG with the counterfactual prompt produced by PAR, while keeping the generation model and sampling process completely unchanged. This setting therefore evaluates whether more refined prompts alone can account for the observed improvements. The results show that relying solely on prompt refinement yields only limited improvement over the Wan2.1 baseline (from 0.40 to 0.43), which is far smaller than the gain achieved when incorporating the full SDG framework (from 0.40 to 0.50). These results demonstrate that prompt design alone is insufficient, and that the SDG mechanism plays the decisive role in improving physical plausibility by directly influencing the denoising trajectory.
> >
> > **(2) On the Role of Language-Guided Conditioning in Current Video Models**
> >
> > Regarding the reviewer’s broader point on whether a “general-purpose world simulator” should internally represent physical laws without relying on language-guided adjustments, we agree that this remains an important long-term goal. However, even recent state-of-the-art systems such as Sora 2 [1] require vision–language models to construct high-fidelity, physically grounded prompts to enhance physical correctness in text-conditioned diffusion models. This suggests that language-guided physical conditioning remains an essential and practical component in current-generation video models. In this context, our SDG framework provides an effective inference-time mechanism that significantly strengthens a model’s physical plausibility without retraining, and does so in a way that cannot be explained by improved prompt-following alone.
> >
> > Reference:
> >
> > [1] Peng X, Zheng Z, Shen C, et al. Open-sora 2.0: Training a commercial-level video generation model in \$200 k[J]. arXiv preprint arXiv:2503.09642, 2025.
> >
> > # Response to Q1
> >
> > Thank you for your question. The implementation is currently being organized and refined. If the paper is accepted, we plan to release the code as open source for the research community to use.

---

### Official Review · Reviewer_qTUC · 2025-10-29

**Soundness:** 3
**Presentation:** 3
**Contribution:** 3
**Rating:** 4
**Confidence:** 5

**Summary:**

The paper proposed an interesting training-free method for enhancing video generation model's obedience to physical rules. The method is similar with negative prompting in image generation, but with 3 key differences:
1) the negative prompt is rewritten from the user prompt by an LLM, with explicitly physical implausible descriptions instead.
2) authors find that at early stage of denoising, the magnitude of  $∆_t = (ϵ^+_t - ϵ^-_t)$ is too small, thus they propose to use the direction of  $∆_t$ instead
3) different from negative prompting that both $ϵ^+$ and $ϵ^-$ are estimated on the same $x_t$, authors propose to always keep both the positive denoising trajectory and the negative denoising trajectory, as specified in line 277~288.

**Strengths:**

The proposed method is totally training-free, and can be plugged into any video generation model's inference process. Possible gain is moderate according to results presented in Table 1&2.

**Weaknesses:**

Authors only present metrics about physics obedience in the paper, it would be much better if there were also analysis and report about method's effect on video's visual quality as well as additional inference cost.

**Questions:**

1. My major concern about the method is about the "Trajectory-Decoupled Denoising" part in page 6. With *decoupled trajectory*, original branch $x_t^+$ and counterfactual branch $x_t^-$ are seperately inferenced (starting from the same intial noise I assume). My question is, when it comes to the later stage of denoising, $x_t^+$ would be remarkably different from $x_t^-$, with $∆_t$ calculated from such different trajectories, won't this compromise the visual quality of the generated video?
2. For the proposed method, the negative prompt is rewritten from the user prompt with explicit physics implausible description, that means, only ONE possible negative result could be mitigated by this method. So my question is, based on the current status of the method, how can we expand the effect to avoid more physics implausible generations?

---

> ### Author Response · Authors · 2025-11-26
>
> Thank you for the reviewer’s constructive and insightful comments. Below we address each concern in detail.
>
> # Response to W1 & Q1
>
>
> **(1) On Trajectory-Decoupled Denoising**
>
> We would like to clarify the role of the guidance term in our Trajectory-Decoupled Denoising.
> In our formulation, the only information from the counterfactual branch that influences the forward generation branch is the normalized contrastive term:
>
> \begin{equation}
>     \frac{\epsilon^{+} - \epsilon^{-}}{\|\epsilon^{+} - \epsilon^{-}\| + \varepsilon}
> \end{equation}
>
> This term is a purely directional vector in latent space. It does not encode spatial structure, image textures, or semantic content, and therefore cannot inject any visual artifacts into the generation. Its effect is limited to modifying the directional component of the denoising update, ensuring that the suppression of implausible trajectories is applied consistently. Since the term contains no pixel-level or texture-level information, the visual fidelity of the generated video is not compromised even when the two trajectories diverge in later steps.
>
> **(2) Video quality**
>
> To address this concern empirically, we further evaluated video quality using six reference-free metrics from VBench [1] and the CLIP Similarity Score [2]. Both the Wan2.1-14B baseline and our model were tested under identical settings. The results are shown below:
>
> **Video Quality Comparison**
>
> | Models                       | CLIPSIM | Image Quality (VBench) | Subject Consistency (VBench) | Background Consistency (VBench) | Motion Smoothness (VBench) | Dynamic Degree (VBench) | Aesthetic Quality (VBench) |
> |-----------------------------|:-------:|:-----------------------:|:------------------------------:|:--------------------------------:|:---------------------------:|:-------------------------:|:----------------------------:|
> | Wan2.1-14B (baseline)       | 0.27    | 0.51                    | 0.82                           | 0.91                             | 0.98                        | 0.39                    | 0.48                         |
> | Wan2.1-14B + Ours       | 0.27.  | 0.50                | 0.81                       | 0.91                        | 0.98                    | 0.41                | 0.48                     |
>
> The results demonstrate that our model maintains video quality that is comparable to the baseline across all metrics. There is no evidence of degradation, indicating that the physical-plausibility improvements introduced by SDG do not come at the expense of video quality.
>
> Reference:
>
> [1] Huang Z, He Y, Yu J, et al. VBench: Comprehensive benchmark suite for video generative models[C]//Proceedings of the IEEE/CVF Conference on Computer Vision and Pattern Recognition. 2024: 21807--21818.
>
> [2] Ramesh A, Dhariwal P, Nichol A, et al. Hierarchical text-conditional image generation with clip latents[J]. arXiv preprint arXiv:2204.06125, 2022, 1(2): 3.
>
> # Response to Q2
>
> We thank the reviewer for this insightful question. To directly address whether incorporating multiple counterfactual prompts could further improve physical plausibility, we conducted an additional experiment where the counterfactual branch is extended from one to two counterfactual prompts. The modified guidance takes the following form:
>
> \begin{equation}
> \epsilon^+ = \epsilon_{\theta}(x_t^+, c(p_+), t) + w \cdot \big(\epsilon_{\theta}(x_t^+, c(p_+), t)-\epsilon_{\theta}(x_t^+, \emptyset, t)\big),
> \end{equation}
> \begin{equation}
> \epsilon_1^- = \epsilon_{\theta}(x_t^+, c(p_1-), t) + w \cdot \big(\epsilon_{\theta}(x_t^-, c(p_1-), t) - \epsilon_{\theta}(x_t^-, \emptyset, t)\big).
> \end{equation}
> \begin{equation}
> \epsilon_2^- = \epsilon_{\theta}(x_t^-, c(p_2-), t) + w \cdot \big(\epsilon_{\theta}(x_t^-, c(p_2-), t) - \epsilon_{\theta}(x_t^-, \emptyset, t)\big).
> \end{equation}
>
>
> and the counterfactual guidance used for sampling was taken as the average
> \begin{equation}
> \epsilon^- \=\ \frac{\epsilon^{-}_1 + \epsilon^{-}_2}{2}.
> \end{equation}
>
> The quantitative result on PhyGenBench is given in the following table:
>
> **Effect of Using One vs. Two Counterfactual Prompts in SDG**
>
> | **Model**                                      | **PhyGenBench (Average PCA)** |
> |------------------------------------------------|:------------------------------:|
> | Wan2.1 (baseline)                              | 0.40                           |
> | **Wan2.1 + SDG (single counterfactual)**        | **0.50**                       |
> | Wan2.1 + SDG (two counterfactual prompts)       | 0.46                           |
>
> These experiments indicate that the current formulation—using one targeted, LLM-derived counterfactual prompt is already the most effective design under the SDG framework. Although the idea of covering multiple implausible outcomes is appealing, naive multi-counterfactual averaging does not improve physical plausibility and may even degrade it. We will explore more principled multi-negative formulation in future work.

---

### Official Review · Reviewer_7HDW · 2025-10-31

**Soundness:** 3
**Presentation:** 3
**Contribution:** 4
**Rating:** 6
**Confidence:** 4

**Summary:**

Summary:
This paper addresses a key limitation of existing diffusion-based video generation models: while these models can produce visually realistic content, they often generate physically implausible results that violate fundamental physical laws. To solve this problem, the paper proposes a training-free framework that enhances physical plausibility during the inference phase.

Contributions:
（1）Training-Free Physics-Aware Paradigm: The framework eliminates the high costs of retraining or fine-tuning, a major limitation of prior physics-aware video generation methods. It acts as a plug-and-play solution that can be integrated with existing diffusion models, filling the gap in inference-time physical control for video generation.
（2）Reasoning-Driven Counterfactual Construction: Unlike generic negative prompts that often lead to irrelevant violations, the Physics-Aware Reasoning pipeline leverages LLMs to generate targeted counterfactuals. These counterfactuals directly violate specific physical laws while preserving the original scene’s entities and context, ensuring that suppression signals are tightly aligned with physical principles.
（3）Innovative Synchronized Decoupled Guidance: The Synchronized Decoupled Guidance strategy addresses the inherent flaws of traditional negative prompting through two complementary designs. Synchronized directional normalization enables early-stage suppression by focusing on the direction of suppression signals (rather than magnitude), while trajectory-decoupled denoising eliminates cumulative bias by evolving parallel latent trajectories for user prompts and counterfactual prompts. Ablation studies confirm that both designs are critical for improving physical plausibility.
（4）Broad Generalization and Practical Utility: The framework achieves consistent performance improvements across four key physical domains (mechanics, optics, thermodynamics, and material interactions) when applied to two representative backbones. It also remains competitive with physics-aware models that require additional training. By preserving photorealism and inference efficiency, the framework supports practical applications such as scientific visualization and realistic content generation.

**Strengths:**

Strengths:
（1）The paper stands out for its creative reimagining of physics-aware video generation, addressing critical gaps in prior work:
1）It introduces atraining-free, inference-only framework—a departure from existing methods (e.g., PhyT2V, WISA) that require costly retraining or fine-tuning. This removes practical barriers to adopting physics enhancement for real-world use.
2）Instead of generic negative prompts, it uses aLLM-powered Physics-Aware Reasoning (PAR) pipelineto generate targeted counterfactuals. These prompts violate specific physical laws while preserving scene entities, turning negative prompting into a precise physics-guided tool rather than a vague avoidance mechanism.

（2）The work maintains high rigor through sound methodology and comprehensive validation:
1）Methodological soundness: PAR’s LLM instruction template (with clear rules for entity analysis and counterfactual construction) ensures consistency. SDG’s design is mathematically grounded in diffusion model principles, avoiding arbitrary choices.
2）Comprehensive experiments: It validates across two backbones (CogVideoX-5B, Wan2.1-14B) to prove generality, four physical domains (mechanics, optics, thermodynamics, material interactions) to show cross-domain effectiveness, and rigorous ablations to confirm that PAR and SDG’s two components are all necessary.

（3）The paper is well-organized and accessible, ensuring complex ideas are understandable:
1）Logical structure: It follows a tight "problem→method→result" flow—Introduction frames limitations of existing models/negative prompting; Methodology links each component (PAR, SDG) to a specific flaw it solves; Experiments organize results by benchmark, domain, and ablation for clarity.
2）Technical transparency: Complex concepts (e.g., SDG’s designs) are explained with equations and plain language, and PAR’s workflow is visualized. Key terms (e.g., "counterfactual prompt") are defined consistently, avoiding jargon overload.

**Weaknesses:**

Weaknesses:
（1）Over-Reliance on LLM Physics Accuracy Without Mitigation
The framework’s Physics-Aware Reasoning (PAR) depends entirely on LLMs to generate valid counterfactual prompts, yet it ignores LLMs’ limitations in niche physical domains (e.g., non-Newtonian fluids, electromagnetism). No experiments test PAR’s ability to handle such scenarios, nor is there a check for LLM-generated counterfactual errors (e.g., misstating viscosity effects).
（2）No SDG Hyperparameter Sensitivity Analysis
SDG uses hyperparameters λ (directional normalization scale) and w (guidance strength), but their impact across domains/backbones is untested. A fixed λ may work for mechanics but fail for subtle optics (e.g., light scattering), harming “plug-and-play” utility.
（3）No Validation on Ambiguous/Low-Quality Prompts
Experiments use clear prompts (e.g., “A tennis ball thrown to the ground”) but ignore real-world vague prompts (e.g., “Something falling in water”) or typos. Ambiguity forces LLMs to make arbitrary assumptions, leading to misaligned counterfactuals.
（4）Missing Comparisons to Inference-Time Baselines
The paper only compares to trained physics models (e.g., PhyT2V) but omits recent inference-time methods (e.g.,Think Before You Diffuse,VLiPP) that also avoid retraining. This hides how SDG stacks up to peers.

**Questions:**

Questions:
（1）Questions About LLM Reliability in PAR
You use LLMs for Physics-Aware Reasoning to generate counterfactual prompts. Do you have data on how often LLMs produce inaccurate counterfactuals related to physical laws? How much do these inaccuracies reduce SDG’s ability to improve physical plausibility? A response will clarify if LLM dependence risks PAR’s effectiveness.
（2）Questions About SDG Hyperparameters
SDG uses λ and w as key hyperparameters. Do these values perform consistently across all four physical domains tested? If not, what hyperparameter settings work best for each domain? This will resolve confusion about SDG’s plug-and-play adaptability.
（3）Questions About Ambiguous Prompt Handling
Your experiments use clear prompts. Have you tested how PAR and SDG perform with vague user prompts? Does ambiguity lead to misaligned counterfactuals and worse physical plausibility scores? Answers will show the framework’s real-world utility.
（4）Suggestions for Inference-Time Baselines
You compare only to trained physics-aware models. Add comparisons to recent inference-time physics-aware methods that also avoid retraining. This will better position SDG against similar state-of-the-art approaches.
（5）Suggestions for Niche Physics Validation
You test PAR on common physics domains. Validate it on niche domains too, and add a lightweight physics verifier to filter inaccurate counterfactuals. This will improve the framework’s robustness across more scenarios.

---

> ### Author Response · Authors · 2025-11-26
>
> Thank you for the reviewer’s constructive and insightful comments. Below we address each concern in detail.
>
> # Response to W1 & Q1 & Q5
>
>
> **On the concern about potential over-reliance on LLM-generated physics reasoning**
>
> Our experiments are conducted on two widely used benchmarks, PhyGenBench and VideoPhy, and we use the exactly the same prompts from the benchmarks. Physical plausibility is evaluated using the official auto evaluators provided by the benchmarks. To clarify the scope, we summarize in the table below the major physical processes and entities represented in these benchmarks. These categories correspond to widely studied and canonical physical phenomena such as Newtonian mechanics, basic thermodynamics, everyday fluid behavior, and classical optics. They also reflect the dominant physical patterns present in large-scale video datasets, which currently define the standard evaluation space in physics-aware video generation.
> At present, there is no established benchmark for niche or highly specialized physical domains such as non-Newtonian fluids or electromagnetism. This makes systematic and comparable evaluation in these domains infeasible. For this reason, our main experiments focus on physical domains where validated evaluation protocols exist, ensuring scientific fairness and reproducibility.
>
>
> **On the capability of PAR in non-standard or niche physical settings**
>
> To more directly address the reviewer’s concern, we constructed several in-house prompts that target niche physical scenarios and evaluated the PAR module under these settings. The results in the following table show that the LLM is able to correctly identify the underlying physical category of each niche-domain prompt. Moreover, for each case, the LLM provides a detailed and physically coherent reasoning process, explicitly describing the roles of the relevant materials, forces, and state transitions in the dynamic evolution of the system. This indicates that the LLM demonstrates reliable physics-aware reasoning within PAR, even for prompts that lie outside the standard benchmark domains.

---

> ### Author Response · Authors · 2025-11-26
>
> # Response to W1 & Q1 & Q5
>
>
> **Niche Physical Prompts and PAR Results**
>
> | **Original Prompt (Physical Domain)** | **PAR Result** |
> |--------------------------------------|----------------|
> | A bowl of cornstarch mixed with water is tapped repeatedly by a hand on a kitchen table. *(Non-Newtonian fluid)* | As the hand taps the mixture, the cornstarch-water fluid instantly transitions to a solid state and remains rigid without reverting to a liquid form, regardless of the removal of stress. |
>
> ### **Physics Reasoning Process**
>
> (i) The specialized physical entity involved in this phenomenon is a **non-Newtonian shear-thickening fluid**, created by mixing cornstarch with water. This mixture exhibits unique rheological behavior where viscosity increases with applied shear rate.
> (ii) As the hand taps the surface, the stress causes particles to lock into a rigid, jammed state due to rapid packing of suspended particles. Upon removal of stress, the fluid relaxes back to a liquid-like state.
> (iii) Governed by shear-thickening behavior and microstructural particle dynamics that respond to high shear conditions.
>
> ---
> | **Original Prompt (Physical Domain)** | **PAR Result** |
> |--------------------------------------|----------------|
> | A small bar magnet is brought close to a clear glass bottle filled with dark ferrofluid. *(Magnetism)* | As the bar magnet approaches, the ferrofluid inside the bottle gradually becomes transparent, losing its opacity entirely without changing its physical structure or particle alignment. |
>
> ### **Physics Reasoning Process**
>
> (i) Entities: **bar magnet + ferrofluid**, a colloidal suspension of magnetic nanoparticles.
> (ii) As the magnet approaches, ferrofluid forms spike-like or chain-like structures along magnetic field lines due to particle magnetization.
> (iii) Controlled by magnetohydrodynamics (MHD) — balancing magnetic forces and viscous forces. Particles align with field lines, forming visible spike patterns.
>
> ---
>
> | **Original Prompt (Physical Domain)** | **PAR Result** |
> |--------------------------------------|----------------|
> | A spoon is slowly dragged through a cup of thick shampoo resting on a bathroom sink. *(Non-Newtonian fluid)* | The spoon is slowly dragged through the shampoo, causing the fluid to thicken and resist movement more as the shear rate increases, eventually becoming completely solid around the spoon. |
>
> ### **Physics Reasoning Process**
> Shampoo exhibits **shear-thinning rheology**.  Initially, shampoo resists motion due to high viscosity at low shear.  As the spoon moves, viscosity decreases, making the fluid flow more easily.  Polymer chains or micelles align with shear, reducing internal resistance.  Once motion stops, the shampoo returns to its original viscosity.
>
> ---
> | **Original Prompt (Physical Domain)** | **PAR Result** |
> |--------------------------------------|----------------|
> | A plastic comb rubbed on dry hair is gradually moved toward a thin stream of water flowing from a faucet. *(Electrostatics)* | The stream unexpectedly bends *away* from the comb, appearing to repel rather than attract, contradicting electrostatic principles. |
>
> ### **Physics Reasoning Process**
> A charged plastic comb **(via triboelectric charging)** polarizes water molecules.  Water dipoles align, making the stream bend toward the comb (attraction), not away.  Underlying interactions involve Coulomb forces and Maxwell’s equations governing polarization and electrostatic attraction.
>
> ---
> | **Original Prompt (Physical Domain)** | **PAR Result** |
> |--------------------------------------|----------------|
> | A strip of stretchy rubber is quickly stretched and released several times between hands. *(Viscoelasticity)* | The rubber strip instantaneously returns to its original shape without any delay or energy dissipation each time it is released. |
>
> ### **Physics Reasoning Process**
>
> Rubber is a **viscoelastic polymer**.  When stretched, polymer chains elongate and store elastic energy.  When released, chains retract but dissipate energy through internal friction—visible as hysteresis.  Described by Maxwell or Kelvin-Voigt viscoelastic models.
>
> ---
> | **Original Prompt (Physical Domain)** | **PAR Result** |
> |--------------------------------------|----------------|
> | A small droplet of oil is carefully placed on the surface of water in a transparent glass bowl. *(Fluid mechanics)* | The droplet of oil spontaneously mixes with the water, vanishing without any external agitation or surfactant presence. |
>
> ### **Physics Reasoning Process**
> This phenomenon involves **interfacial tension + immiscibility**.
> (i) Oil (non-polar) vs water (polar) remain separate phases.
> (ii) High interfacial tension keeps the oil droplet intact; it may spread slightly but will not mix.
> (iii) Governed by Young–Laplace equation and spreading coefficients. Equilibrium depends on energetic balance among oil–water–air interfaces.
>
> ---

---

> ### Author Response · Authors · 2025-11-26
>
> # Response to W2 & Q2
>
> We conducted ablation experiments on both key hyperparameters: the guidance strength $w$ and the directional normalization scaling factor $\lambda$. The results are reported in the following tables:
>
> **Ablation on Guidance Scale**
>
> | Guidance Scale | Wan2.1-14B (Baseline) | Wan2.1-14B + Ours |
> |----------------|:----------------------:|:------------------:|
> | \(w = 5\)      | 0.40                   | **0.50**           |
> | \(w = 7\)      | 0.44                   | 0.43               |
> | \(w = 9\)      | 0.43                   | 0.43               |
> | \(w = 11\)     | 0.43                   | 0.45               |
>
> **Ablation on Scaling Factor**
>
> | **Scaling Factor** | **Wan2.1-14B + Ours** |
> |-------------------|:----------------------:|
> | $\lambda = 20$    | 0.46                   |
> | $\lambda = 30$    | **0.50**               |
> | $\lambda = 40$    | 0.43                   |
>
> The results show that our method achieves the largest improvement at $w$ = 5, clearly surpassing the baseline. Across all tested values, SDG never underperforms the baseline.
>
> Similarly, SDG remains stable across a wide range of $\lambda$. The best performance is obtained at $\lambda$ = 30, and even suboptimal values exhibit no catastrophic drop, demonstrating that SDG does not require fine-grained hyperparameter tuning.
>
>
> # Response to W3 & Q3
>
> **(1) The choice of prompts**
>
> We would like to clarify that the choice of prompts is determined by the evaluation protocol of the field. To ensure fair comparison with prior work and to rigorously assess improvements in physical plausibility, we follow the two community-standard benchmarks, PhyGenBench and VideoPhy, and use exactly the same prompts provided by these benchmarks. These benchmarks supply dedicated auto evaluators that assess the physical plausibility of generated videos. Importantly, these evaluators are trained and validated specifically on the official prompts, and their papers report a high correlation between evaluator judgments and human evaluations.
> To enable consistent and interpretable assessment of physical plausibility, the official prompts are intentionally written in a clear and well-specified manner, including explicit physical laws, object properties, and expected state transitions. This clarity is necessary for generating unambiguous ground-truth expectations and for training the auto evaluators to reliably identify physical violations.
>
> **(2) Why ambiguous or low-quality prompts cannot be reliably evaluated**
>
> Regarding the reviewer’s suggestion to test ambiguous or low-quality prompts such as “Something falling in water,” we agree that such cases are interesting, but currently there is no established benchmark or evaluator that can provide reliable ground-truth judgments for ambiguous prompts. Assessing physical consistency requires specific physical attributes of the entities involved. For example, when “something” falls into water, the correct physical behavior depends on material density, buoyancy, geometry, and interaction forces. Even human observers cannot determine the physically expected outcome without knowing what the “something” is; a feather, a steel ball, and a rubber duck produce fundamentally different dynamics.
> Because no deterministic ground-truth trajectory exists for ambiguous prompts, and because no benchmark offers validated evaluators for such settings, systematic evaluation of ambiguous prompts is currently infeasible. For this reason, we focus on the officially defined, well-specified prompts where reliable and reproducible assessment of physical plausibility is possible.

---

> > ### Author Response · Authors · 2025-11-26
> >
> > # Response to W4 & Q4
> >
> > We thank the reviewer's suggestion. We have now added Think Before You Diffuse to our quantitative table. As shown in the following table, this method requires fine-tuning a pre-trained diffusion model to achieve its reported performance and therefore is not a training-free method. For fairness, we list it under the “Physics-aware models (trained/fine-tuned)” category.
> >
> > Regarding VLiPP, we note that it is an image-to-video method. Its objective and task setting are therefore not comparable to ours, which is an training-free framework for text-to-video generation models. For this reason, we do not include VLiPP in our comparisons.
> >
> > The updated table including Think Before You Diffuse is provided below.
> >
> > **Quantitative Comparisons on VideoPhy and PhyGenBench**
> >
> > #### *Base Models*
> >
> > | **Model** | Training-Free | SA | PC | PhyGenBench |
> > |-----------|---------------|----|----|--------------|
> > | CogVideoX-2B | -- | -- | -- | 0.39 |
> > | LaVie | -- | -- | -- | 0.43 |
> > | VideoCrafter2 | -- | 0.47 | 0.36 | 0.48 |
> > | Open-Sora | -- | 0.38 | 0.43 | 0.45 |
> > | Vchitect 2.0 | -- | -- | -- | 0.45 |
> > | Cosmos-Diffusion-7B | -- | 0.52 | 0.27 | 0.24 |
> >
> > ---
> >
> > #### *Physics-Aware Models (trained / fine-tuned)*
> >
> > | **Model** | Training-Free | SA | PC | PhyGenBench |
> > |-----------|---------------|----|----|--------------|
> > | Think Before You Diffuse | no | -- | -- | 0.54 |
> > | PhyT2V (Round 4) | no | 0.59 | 0.42 | 0.42 |
> > | DiffPhy | no | -- | -- | 0.54 |
> > | VideoREPA-5B | no | 0.72 | 0.40 | -- |
> > | CogVideoX-5B + WISA | no | 0.67 | 0.38 | 0.43 |
> >
> > ---
> >
> > #### *Ours (training-free, inference-time) applied to two baselines*
> >
> > | **Model** | Training-Free | SA | PC | PhyGenBench |
> > |-----------|---------------|----|----|--------------|
> > | CogVideoX-5B | -- | 0.48 | 0.39 | 0.47 |
> > | **CogVideoX-5B + Ours** | **yes** | **0.49** | **0.40** | **0.49** |
> > | Wan2.1-14B | -- | 0.49 | **0.35** | 0.40 |
> > | **Wan2.1-14B + Ours** | **yes** | **0.52** | **0.35** | **0.50** |
> >
> > ---

---

### Official Review · Reviewer_qJPD · 2025-11-03

**Soundness:** 3
**Presentation:** 2
**Contribution:** 3
**Rating:** 4
**Confidence:** 4

**Summary:**

This paper proposes a training free framework to enhance video diffusion models' physical plausibility. In detail, the pipeline first generate counterfactual prompts that describe physics-violating behaviors; then they propose a Synchronized Decoupled Guidance (SDG) strategy to mitigate implausible contents. This framework is highly related to negative prompt, but the key differences are: 1) it has a better designed negative prompts 2) It keeps two stream instead of one, the negative branch is denoised independently 3) It use only the direction instead of the mode. These changes are aiming to fix two gaps in negative prompting: 1) the suppression effect is weak at the beginning 2) the two branch shares same x_t, which causes cumulative bias.

**Strengths:**

1. This method is easy to understand, easy to implement, and plug-and-play, the authors provide pretty good ablations in Table 3, helps me to better understand the method.
2. Happy to see the authors tested more than one video models

**Weaknesses:**

1. maintaining two branchs almost doubles the cost, while in Table 3, the w/o Trajectory-decoupled denoising is getting 0.48, which is super close to 0.48. seems this does not contribute too much to the final results.
2. I checked the supp videos, it seems both baseline videos and SDG videos have some artifacts at the first several frames, this should not happens in Wan, I suspect there might be some implementation errors
3. In the supp videos, I found several cases have significantly over saturated generation, for example, the "A silver spoon is slowly inserted into a glass of crystal-clear water, revealing the fascinating visual changes and reflections as the spoon interacts with the liquid.", this is related to the guidance strength of 30 used in SDG, I highly suspect this will sacrafice FVD score, or video quality, the authors should also provide quality-wise eval.
4. Lacks the details of guidance scale for CFG baseline, also authors should provide a sweep over different guidance scale for CFG and SDG, for both physical benchmarks and video quality metrics.

**Questions:**

1. What's the guidance strength of cfg baseline, I am afraid the gap is mainly coming from a lower guidance scale for cfg. The authors should provide evidences to clarify.

---

> ### Author Response · Authors · 2025-11-26
>
> # Response to W1
>
> We appreciate the reviewer for this insightful question and would like to clarify a misconception. The additional computational cost corresponds to the performance improvement from the “w/o Synchronized Decoupled Guidance” variant (0.43) to the full version (0.50), not from the “w/o Trajectory-Decoupled Denoising” variant (0.48) as mentioned in the comment. Importantly, Trajectory-Decoupled Denoising (TDD) does not introduce extra computational overhead: its inference cost is identical to the full model, thus the 0.48 to 0.50 gain arises purely from its algorithmic design rather than increased computation.
>
> Below, we would like to address this in more detail by explaining (1) why our inference overhead is modest compared to alternative designs for such a significant performance gain (from 0.43 to 0.50), (2) the purpose of the “w/o Trajectory-decoupled denoising (TDD)” ablation, and how it differs from the other ablated versions.
>
> **(1) On computational cost**
>
> While the Synchronized Decoupled Guidance (SDG) framework introduces an additional inference branch, the overhead remains modest compared to the immense training and data-curation costs of modern video diffusion models. Our method is entirely training-free, requires no model modification, and can be applied directly to any pretrained backbone. Therefore, the trade-off between the doubled inference-time cost and the notable performance gain (0.43 to 0.50) is practical and efficient.
>
>  **(2) On the purpose of the “w/o Trajectory-decoupled denoising (TDD)” ablation**
>
> This ablation is designed to isolate the effect of trajectory coupling rather than assess the entire SDG framework, as TDD is only one component of SDG. In this setting, we intentionally reintroduce coupling between the forward and counterfactual branches by modifying Eqs. (11)–(12) as:
>
> \begin{equation}
> \epsilon^+ = \epsilon_{\theta}(x_t^+, c(p_+), t) + w \cdot \big(\epsilon_{\theta}(x_t^+, c(p_+), t)-\epsilon_{\theta}(x_t^+, \emptyset, t)\big),
> \end{equation}
>
> \begin{equation}
> \epsilon^- = \epsilon _ {\theta}(\textcolor{red}{\mathbf{x _ t^+}}, c(p _ -), t) + w \cdot \big(\epsilon _ {\theta}(x _ t^-, c(p _ -), t) - \epsilon _ {\theta}(x _ t^-, \emptyset, t)\big).
> \end{equation}
>
> Thus, the difference between the “w/o TDD” and full versions reflects only the impact of trajectory coupling, while the true benefit of SDG is captured by the comparison between “w/o SDG” (0.43) and the full version (0.50).
>
> Because TDD alone evaluates only the decoupling mechanism, we additionally include a new ablation to test the importance of maintaining two synchronized trajectories:
>
> \begin{equation}
> \epsilon^+ = \epsilon_{\theta}(x_t^+, c(p_+), t) + w \cdot \big(\epsilon_{\theta}(x_t^+, c(p_+), t)-\epsilon_{\theta}(x_t^+,\emptyset,t)\big),
> \end{equation}
> \begin{equation}
> \epsilon^- = \epsilon_{\theta}(\textcolor{red}{\mathbf{x_t^+}}, c(p_-), t)+ w \cdot \big(\epsilon_{\theta}(\textcolor{red}{\mathbf{x_t^+}}, c(p_-), t)- \epsilon_{\theta}(\textcolor{red}{\mathbf{x_t^+}}, \emptyset, t)\big).
> \end{equation}
>
> Because removing the dual-branch structure reintroduces inherent coupling, the performance drops substantially from 0.50 → 0.43, confirming that the synchronized two-trajectory structure is essential.
>
> The updated ablation table is:
>
> **Ablation Study**
>
> | Model                                      | Average |
> |--------------------------------------------|:-------:|
> | Wan2.1-14B                                 | 0.40    |
> | w/o Synchronized decoupled guidance        | 0.43    |
> | w/o Synchronized trajectory                | 0.43    |
> | w/o Synchronized directional normalization | 0.47    |
> | w/o Trajectory-decoupled denoising         | 0.48    |
> | w/o Physics-aware reasoning                | 0.47    |
> | **Full version (Ours)**                    | **0.50** |
>
> We appreciate the reviewer’s feedback and have added explicit definitions for each ablation variant in Sec. 3.3 of the revised paper, with all changes highlighted in blue for clarity.

---

> > ### Author Response · Authors · 2025-11-26
> >
> > # Response to W2
> >
> > We thank the reviewer for carefully examining the supplementary videos. After re-checking the material, we confirm that a small number of videos do exhibit noticeable noise in the first several frames. Importantly, this issue is not caused by an implementation error.
> >
> > In our experiments, we adopted one of the lightweight configurations used in CogVideoX (FPS = 8, number of frames = 25) to ensure fair comparison across all methods. While this reduced-cost setting is default for large-scale evaluation, it can introduce minor temporal artifacts in the early frames of Wan2.1 outputs for visualisation. These artifacts do not affect our conclusions regarding the improvements in physical plausibility.
> >
> > To address this concern, we regenerated representative samples for Wan using the standard Wan2.1 configuration (FPS = 16, number of frames = 81). Under the default setting, the noise artifacts disappear completely. We have included several high-quality comparison videos in the supplementary material for the reviewer’s reference.
> >
> > # Response to W3
> >
> > **(1) On the oversaturation observed in supplementary videos**
> >
> > We thank the reviewer for pointing out the oversaturation in several supplementary examples. This issue is not caused by the guidance strength used in SDG. Instead, it primarily results from the fact that, for both the baseline and our method, we intentionally removed the negative prompt used in CFG (e.g., bright tones, overexposed, blurred details, JPEG artifacts, low quality, deformed limbs, etc., in Wan2.1 baseline).
> >
> > We removed these negative prompts because our goal was to isolate and analyze the contribution of Physics-aware Reasoning (PAR) and Synchronized Decoupled Guidance (SDG) to physical plausibility. Retaining negative prompts would introduce additional suppression effects during denoising, making it difficult to attribute improvements specifically to SDG.
> >
> > However, removing negative prompts can indeed lead to mild oversaturation in both the baseline and SDG outputs. To confirm this, we regenerated videos using the original CFG negative prompts. As shown in the supplementary material, reintroducing negative prompts effectively mitigates oversaturation for both models, validating that the issue arises from prompt configuration rather than SDG itself.
> >
> > **(2) On concerns regarding video-quality degradation and the need for quality-wise evaluation**
> >
> > Regarding video quality, we would like to clarify that FVD is not applicable in our evaluation setting because it requires real reference videos, which neither PhyGenBench nor VideoPhy provides. As a result, using FVD would not yield meaningful or interpretable results.
> >
> > Instead, we adopt six reference-free video-quality metrics from VBench [1], which are specifically designed for arbitrary text-to-video outputs:
> > subject consistency, background consistency, motion smoothness, dynamic degree, aesthetic quality, and imaging quality.
> > We also report the CLIP Similarity Score [2] to measure semantic alignment with the input prompt.
> >
> > Both the Wan2.1-14B baseline and our SDG-enhanced model are evaluated under identical settings. The results are shown below:
> >
> > **Video Quality Comparison**
> >
> > | Models                       | CLIPSIM | Image Quality (VBench) | Subject Consistency (VBench) | Background Consistency (VBench) | Motion Smoothness (VBench) | Dynamic Degree (VBench) | Aesthetic Quality (VBench) |
> > |-----------------------------|:-------:|:-----------------------:|:------------------------------:|:--------------------------------:|:---------------------------:|:-------------------------:|:----------------------------:|
> > | Wan2.1-14B (baseline)       | 0.27    | 0.51                    | 0.82                           | 0.91                             | 0.98                        | 0.39                    | 0.48                         |
> > | Wan2.1-14B + Ours       | 0.27 | 0.50                | 0.81                       | 0.91                         | 0.98                    | 0.41                | 0.48                     |
> >
> > The results demonstrate that our model maintains video quality that is comparable to the baseline across all metrics. There is no evidence of degradation, indicating that the physical-plausibility improvements introduced by SDG do not come at the expense of video quality.
> >
> > Reference:
> >
> > [1] Huang Z, He Y, Yu J, et al. VBench: Comprehensive benchmark suite for video generative models[C]//Proceedings of the IEEE/CVF Conference on Computer Vision and Pattern Recognition. 2024: 21807--21818.
> >
> > [2] Ramesh A, Dhariwal P, Nichol A, et al. Hierarchical text-conditional image generation with clip latents[J]. arXiv preprint arXiv:2204.06125, 2022, 1(2): 3.

---

> ### Author Response · Authors · 2025-11-26
>
> # Response to W4 & Q1
>
> We have supplemented our paper with a detailed evaluation of the guidance scale for both the CFG baseline and our SDG method. Specifically, we perform a sweep over multiple values of the guidance strength w and report the resulting Physical Commonsense Alignment (PCA) scores on PhyGenBench for both Wan2.1-14B and Wan2.1-14B + SDG. The results are summarized in the following table:
>
> **Ablation on Guidance Scale**
>
> | Guidance Scale | Wan2.1-14B (Baseline) | Wan2.1-14B + Ours |
> |----------------|:----------------------:|:------------------:|
> | \(w = 5\)      | 0.40                   | **0.50**           |
> | \(w = 7\)      | 0.44                   | 0.43               |
> | \(w = 9\)      | 0.43                   | 0.43               |
> | \(w = 11\)     | 0.43                   | 0.45               |
>
> The experiment shows that our method achieves the strongest improvement at $w$ = 5, outperforming the baseline by a clear margin.
>
> In addition, we conducted experiments to evaluate video quality under different values of the guidance scale $w$. The results show that, as $w$ increases, most video quality metrics consistently degrade, with the exception of Dynamic Degree and CLIP Score. Our method maintains stable performance and does not degrade relative to the baseline.
>
> **Video Quality under Different Guidance Scales**
>
> | Models                         | CLIPSIM | Image Quality (VBench) | Subject Consistency (VBench) | Background Consistency (VBench) | Motion Smoothness (VBench) | Dynamic Degree (VBench) | Aesthetic Quality (VBench) |
> |--------------------------------|:-------:|:-----------------------:|:------------------------------:|:--------------------------------:|:---------------------------:|:-------------------------:|:----------------------------:|
> | Wan2.1 + Ours (w=5)            | 0.27    | 0.51                    | 0.82                           | 0.91                             | 0.98                        | 0.39                     | 0.48                         |
> | Wan2.1 (Baseline) (w=5)        | 0.27    | 0.50                    | 0.81                           | 0.91                             | 0.98                        | 0.41                     | 0.48                         |
> | Wan2.1 (Baseline) (w=7)        | 0.27    | 0.50                    | 0.81                           | 0.91                             | 0.98                        | 0.49                     | 0.48                         |
> | Wan2.1 + Ours (w=7)            | 0.27    | 0.50                    | 0.81                           | 0.91                             | 0.98                        | 0.46                     | 0.49                         |
> | Wan2.1 (Baseline) (w=9)        | 0.26    | 0.51                    | 0.81                           | 0.91                             | 0.98                        | 0.40                     | 0.47                         |
> | Wan2.1 + Ours (w=9)            | 0.27    | 0.50                    | 0.81                           | 0.91                             | 0.98                        | 0.43                     | 0.47                         |
> | Wan2.1 (Baseline) (w=11)       | 0.27    | 0.48                    | 0.79                           | 0.91                             | 0.97                        | 0.47                     | 0.46                         |
> | Wan2.1 + Ours (w=11)           | 0.27    | 0.49                    | 0.80                           | 0.91                             | 0.97                        | 0.47                     | 0.46                         |

---

### Author Response · Authors · 2025-11-29
**General Response to All Reviewers**

We sincerely appreciate all reviewers’ time and constructive feedback. We are glad to find out that the reviewers acknowledge our contributions, including the training-free nature of our framework, the novel physics-aware reasoning, and the effectiveness and generality of SDG. Below we summarize the key updates and clarifications provided in the rebuttal.

### **[Our Contributions]**

We are glad to find out that the reviewers generally acknowledge our contributions:

  - The framework is training-free, plug-and-play, and easy to integrate into existing video diffusion models. (qJPD, 7HDW, qTUC)

  - The paper proposes a novel and well-motivated combination of physics-aware reasoning (PAR) and synchronized decoupled guidance (SDG). (7HDW, TVE5)

  - The counterfactual prompts generated by PAR are more precise and physically grounded than generic negative prompts. (7HDW, TVE5)

  - Results demonstrate consistent improvements across physical domains and backbones, with practical relevance for real-world video generation. (qJPD, 7HDW, TVE5)

  - The paper is clear, well-organized, and supported by solid ablations that help clarify the contribution of each component. (qJPD, 7HDW, qTUC, TVE5)

### **[New Experiments]**
In this rebuttal, we have added more supporting experiments to address reviewers’ concerns:

  - **Expanded SDG ablations:** We add a full ablation including removing directional normalization, trajectory-decoupled denoising (TDD), and both together. Results confirm that SDG is an effective and essential component of the framework.
  - **Hyperparameters:** We sweep the guidance scale w for CFG and SDG, and the scaling factor λ for SDG, showing SDG is stable across a wide range and that our gains are not due to hyperparameter tuning.
  - **Video quality evaluation:** We report six VBench metrics + CLIP-SIM, showing our method preserves video quality while improving physical plausibility.
  - **Niche physics testing for PAR:** We evaluate PAR on specialized domains (non-Newtonian fluids, magnetism, electrostatics, etc.), demonstrating robust and physically coherent reasoning.
  - **Baseline generalization:** Additional results on an additional baseline Vchitect-2.0 show consistent improvement (0.46 → 0.49).

### **[Clarifications]**

We noticed several concerns about our method in the reviews and provide the following clarifications:

  - Early-frame artifacts and oversaturation arise from evaluation settings and missing negative prompts in the controlled setup, not from our proposed method SDG. Regenerated videos eliminate these issues (shown in the updated supplementary material).

  - Guidance scale for CFG baseline is now explicitly reported; SDG’s gains are not due to weaker CFG guidance.

  -  TDD does not harm visual quality; results show comparable video quality metrics.

  -  Using two counterfactual prompts is less effective than one (0.46 vs. 0.50), confirming our design choice.

### **[Summary]**

We believe that the additional experiments and clarifications provided in our rebuttal have directly and comprehensively addressed each of the reviewers’ concerns. Due to platform issues occurring after the rebuttal submission, the reviewers were unfortunately unable to revisit or verify our detailed responses. We hope that this summary clearly conveys to the AC how our rebuttal resolves every point raised across the reviews, and highlights the substantial improvements and clarifications we have made.

---

### Meta-Review · Area_Chair_fCtn · 2025-12-20

**Summary:**

- Method Efficacy & Practicality (qJPD, TVE5): Concerns about the marginal quantitative gains on some benchmarks (e.g., VideoPhy for CogVideoX-5B), unclear practical visual improvement in qualitative examples, and the computational cost of maintaining two denoising trajectories. Questions about whether improvements stem from better prompting or the SDG mechanism itself.

- Technical Robustness & Evaluation (qJPD, 7HDW, qTUC): Lack of video quality evaluation (e.g., FVD, aesthetic metrics). Absence of hyperparameter sensitivity analysis for SDG. Potential visual quality degradation due to trajectory decoupling. Over-reliance on LLMs for PAR without validation in niche physics domains. Missing comparisons to other inference-time baseline methods.

- Clarifications & Implementation (qJPD, TVE5): Questions about early-frame artifacts, oversaturation in videos, the specific guidance scale used for baselines, and the availability of code.

**Reviewer Concerns:**

The authors provided extensive new experiments, but there are still concerns.

- Magnitude of Improvement & Qualitative Clarity (TVE5): While the authors provided explanations (e.g., VideoPhy's evaluator limitations) and pointed to specific improvements in figures, the perceptual impact in some examples may remain subtle. This is a subjective aspect of the contribution.

- Ambiguous Prompts (7HDW): The authors correctly noted that current benchmarks and evaluators do not support ambiguous prompts, making systematic evaluation infeasible. This is a valid point about the field's state, though the concern about real-world applicability remains.

**Reviewer Scores:**

- Reviewer qJPD (Rating: 4): Concerns about cost/benefit, artifacts, quality metrics, and hyperparameters.

- Reviewer 7HDW (Rating: 6): Concerns about niche physics, hyperparameters, and baselines were addressed with new experiments.

- Reviewer qTUC (Rating: 4): Concerns about visual quality and the multi-counterfactual expansion.

- Reviewer TVE5 (Rating: 4): Concerns about generalization, qualitative gains, and the source of improvement.

The authors provide a lot of rebuttal, but the overall score is still lower than the ICLR bar. It is suggested to resubmit with a major revision.

---

### Decision · Program_Chairs · 2026-01-26

Reject